



**Do Loop Current Eddies stimulate productivity in the Gulf of Mexico?**
Pierre Damien[1,2], Julio Sheinbaum[1], Orens Pasqueron de Fommervault[1], Julien Jouanno[3], Lorena
Linacre[4], Olaf Duteil[5]
[1] Departamento de Oceanografía Física, Centro de Investigación Científica y de Educación Superior
de, Ensenada, México,
[2] University of California, Los Angeles, CA
[3] LEGOS, Université de Toulouse, IRD, CNRS, CNES, UPS, Toulouse, France,
[4] Departamento de Oceanografía Biológica, Centro de Investigación Científica y de Educación
Superior de Ensenada, México,
[5] GEOMAR Helmholtz Centre for Ocean Research, Kiel, Germany.
Corresponding author: Pierre Damien (pdamien@ucla.edu)
**Key Points :**
• LCEs trigger a local phytoplankton biomass increase in winter.
• Chlorophyll variability at surface does not reflect the seasonal cycle of the depth-integrated

15       biomass.

• Convective mixing and Ekman pumping are key mechanisms to preferentially supply nutrient

17       toward the euphotic layer in LCEs.





**Abstract**
Surface chlorophyll concentrations inferred from satellite images suggest a strong influence of
the mesoscale activity on biogeochemical variability within the oligotrophic regions of the Gulf of
Mexico (GoM). More specifically, long-living anticyclonic Loop Current Eddies (LCEs) are shed
episodically from the Yucatan Channel and propagate westward. This study addresses the
biogeochemical response of the LCEs to seasonal forcing and show their role in driving phytoplankton
biomass distribution in the GoM. Using an eddy resolving (1/12°) interannual regional simulation
based on the coupled physical-biogeochemical model NEMO-PISCES that yields a realistic
representation of the surface chlorophyll distribution, it is shown that the LCEs foster a large biomass
increase in winter in the upper ocean. The primary production in the LCEs is larger than the average
rate in the surrounding open waters of the GoM. This behavior cannot be directly identified from
surface chlorophyll distribution alone since LCEs are associated with a negative surface chlorophyll
anomaly all year long. This anomalous biomass increase in the LCEs is explained by the mixed-layer
response to winter convective mixing that reaches deeper and nutrient-richer waters.



# I/ Introduction

Historical satellite ocean color observations of the deep waters of the Gulf of Mexico (roughly

delimited by the 200m isobath and from hereafter referred to as GoM open-waters) indicate low surface
chlorophyll concentrations ([CHL]), low biomass and low primary productivity (Müller-Karger et al.,
1991; Biggs and Ressler, 2001; Salmerón-García et al., 2011). The GoM open-waters are mostly
oligotrophic, as confirmed by more recent bio-optical in-situ measurements from autonomous floats
(Green et al., 2014; Pasqueron de Fommervault et al., 2017; Damien et al., 2018). The surface
chlorophyll concentration in the GoM open-waters exhibits a clear seasonal cycle which is primarily
triggered by the seasonal variation of the mixed layer depth (Müller-Karger et al., 2015) and river
discharges (Brokaw et al., 2019). In tandem, the seasonal cycle is strongly modulated by the energetic
mesoscale dynamic activity which shapes the distribution of biogeochemical properties (Biggs and
Ressler, 2001; Pasqueron de Fommervault et al., 2017). This mesoscale activity is dominated by the
large and long-living Loop Currents Eddies (LCEs) which are shed episodically by the Loop Current
(Weisberg and Liu, 2017) and constitute the most energetic circulation features in the GoM
(Sheinbaum et al., 2016; Sturges & Leben, 2000).

Mesoscale activity (see McGuillicuddy et al., 2016 for a review) modulates the phytoplankton

biomass distribution (Siegel et al., 1999; Doney et al., 2003; Gaube et al., 2014; Mahadevan, 2014) and
the ecosystem functioning (McGillicuddy et al., 1998, Oschlies and Garcon, 1998, Garcon et al., 2001).
Specifically, the ability of the mesoscale eddies to enhance vertical fluxes of nutrients is determinant in
sustaining the observed phytoplankton growth rate in oligotrophic regions such as the GoM open-
waters, where the phytoplankton primary production is limited by nutrient availability in the euphotic
layer (McGillicuddy and Robinson 1997; McGillicuddy et al., 1998; Oschlies and Garcon, 1998).





The upward doming of isopycnals in cyclonic eddies and downward depressions in anticyclonic
eddies, also known as "eddy-pumping", occur when the eddies are strengthening (Siegel et al., 1999,
Klein and Lapeyre, 2009) and produce a nutrient vertical transport. This has been historically proposed
as the dominant mechanism controlling the mesoscale biogeochemical variability, as it induces a
reduction of productivity in the anticyclone and an increase in cyclones. This paradigm is however
challenged by observations of enhanced surface chlorophyll concentrations in anticyclonic eddies
(Gaube et al., 2014), particularly during winter (Dufois et al., 2016). As a plausible explanation, eddy-
wind interactions may significantly modulate vertical fluxes through Ekman transport divergence
within the eddies (Martin and Richards, 2001, Gaube et al., 2013, 2015). This mechanism is
responsible for a downwelling in the core of cyclones and an upwelling in the core of anticyclones.
Dufois et al. (2014, 2016) link these observations to a deeper mixed layer in anticylonic eddies. This is
explained by the eddy-driven modulation of the upper ocean stratification which directly affects the
winter convective mixing (He et al., 2017). Observed mixed layers tend to be deeper in anticyclones
than in cyclones (Williams, 1998; Kouketsu et al., 2012) and vertical nutrient fluxes to the euphotic
layer are potentially enhanced in anticyclones during periods prone to convection (e.g. winter in the
GoM). Although some consensus exists on the fundamental role of anticyclonic eddies on the
productivity of oligotrophic ocean regions, large uncertainties remain regarding the relative importance
of the different mechanisms involved in the biogeochemical responses.
Besides, in-situ measurements in oligotrophic regions have shown that the surface [CHL]
variability, observed from ocean color satellite imagery, is not necessarily representative of the total
phytoplankton (carbon) biomass variability in the water column (Siegel et al., 2013; Mignot et al.,
2014). In particular, a surface [CHL] winter increase, may result from physiological mechanisms (i.e.
modification of the ratio of [CHL] to phytoplankton carbon biomass) or from a vertical redistribution





of the phytoplankton (Mayot et al., 2017) rather than from changes in the biomass content. It is not
clear yet which of these hypotheses holds in oligotrophic regions, and more specifically in the GoM
open-waters where this issue has been addressed by in-situ sub-surface [CHL] observations (Pasqueron
de Fommervault et al., 2017). Most of the studies focusing on chlorophyll variability use surface (or
near-surface) [CHL] as a proxy for phytoplankton biomass and interpret a [CHL] increase as an
effective biomass production. Only a few studies considered the vertically integrated responses (Dufois
et al., 2017; Guo et al., 2017; Huang and Xu, 2018) emphasizing the importance of considering the
eddy impact on the subsurface.

The objective of this study is to better understand the role of LCEs in driving [CHL] distribution

and variability within the GoM open-waters. Material and methods used in this study are presented in
section 2. In section 3, the imprint of the LCEs on the surface [CHL] distribution is inferred from
satellite ocean color observations. Since these measurements are confined to the oceanic surface layer
and do not allow access to the vertical properties of LCEs, we complete the analysis with a coupled
physical-biogeochemical simulation (subsections 2 and 3). Particular attention is paid to the validation
of the modeled LCE dynamical structures and surface [CHL] anomalies. In the last section, we propose
to disentangle the mesoscale mechanisms controlling the seasonal cycle of the [CHL] vertical profile in
LCEs. The model also enables to assess both abiotic and biotic processes and physical-biogeochemical
interactions that can be difficult to address with in-situ observations only.
**II/ Material and methods**

**II.1/ The coupled physical-biogeochemical model**



The simulation analyzed in this study (referred as GOLFO12-PISCES) has been described and
compared with observations in Damien et al. (2018). It relies on a physical-biogeochemical coupled
model based on the ocean model NEMO (Nucleus for European Modeling of the Ocean, version 3.6;
Madec, 2016) and the biogeochemical model PISCES (Pelagic Interaction Scheme for Carbon and
Ecosystem Studies; Aumont and Bopp, 2006; Aumont et al., 2015). The model grid covers the GoM
and the western part of the Cayman Sea (Fig 1) with a 1/12° horizontal resolution (~ 8.4 km). This
allows to resolve scales related to the first baroclinic mode, which is of the order of 30-40 km in the
GoM open-waters (e.g., Chelton et al., 1998). The model is forced with realistic open-boundary
conditions, high frequency atmospheric forcing, and freshwater and nutrient-rich discharges from
rivers. The analysis has been performed using 5-day averaged outputs for a period of 5 years from 2002
to 2007. We refer the reader to Damien et al. (2018) for an extended model and numerical setup
descriptions and a careful validation against observations that show the ability of the model to
reproduce the main hydrographic and nutrient vertical distributions in the GoM.

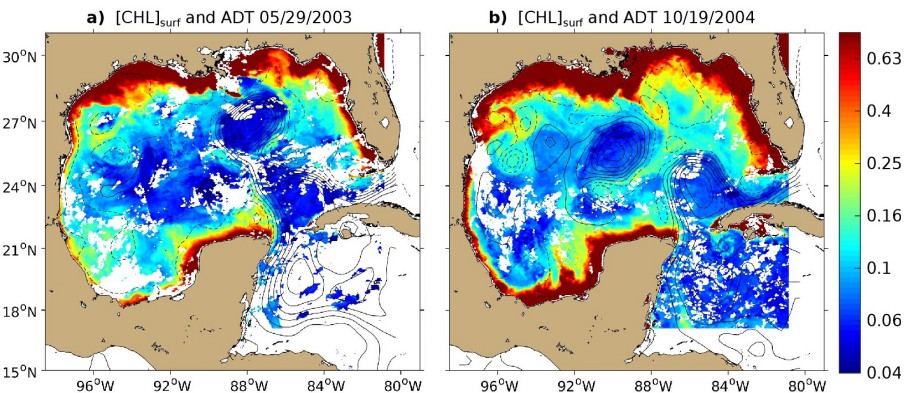

**Figure 1: 8-days composite images of $[CHL]_{surf}$ (in mg·m$^{-3}$) around (a) May 29$^{th}$ 2003 and (b) October 19$^{th}$ 2004 derived from**
**Aqua-MODIS images overlaid with contours of Absolute Dynamic Topography (ADT in m) derived from Aviso images are**
**superimposed. Contour interval is 10cm and ADT values lower than 40cm are shown with dashed curves.**



## II.2/ Observational Data Set Used

Satellite observations are used to evaluate the ability of GOLFO12-PISCES to reproduce the
dynamical and biological signatures associated with LCEs. Surface geostrophic velocities are derived
from a 1/4° multi-satellite merged product of absolute dynamic topography (ADT) provided by
AVISO+ (http://marine.copernicus.eu). Surface chlorophyll concentrations are from the Aqua-MODIS
4 km product (Sathyendranath et al., 2012; http://marine.copernicus.eu) and consist of 8-day
composites from 2003 to 2015.

## II.3/ LCEs detection, tracking and composite construction

In order to track the LCEs, we use the algorithm developed by Nencioli et al. (2010), which has
been extensively employed to track coherent mesoscale eddies (Dong et al., 2012, Ciani et al. 2017,
Zhao et al. 2018) and submesoscale eddies (Damien et al., 2017). It is based on the geometric
organization of the velocity fields, dominated by rotation, that develop around eddy centers. Here, it is
applied to weekly AVISO+ surface geostrophic velocities and GOLFO12-PISCES 5-day averaged
velocities at 20m depth. Since LCEs are surface intensified (Cooper et al., 1990; Forristall et al., 1992;
Sturges and Kenyon, 2008), the choice of a shallow detection depth is expected to maximize the
accuracy. The selection of LCEs is defined using the criteria that eddies have to be shed from the Loop
Current.
In order to assess the [CHL] response to LCE dynamics, eddy-centric horizontal images and
transects of LCEs are used to make composites constructed by averaging modeled variables of the





different LCEs collocated to their center. The transect building procedure involves an axisymmetric
averaging that assumes axis-symmetry of the dynamical structures and no tilting of their rotation axis.
Moreover, we choose not to consider the LCEs formation period and the LCEs destruction period when
reaching the western basin (Lipphardt et al., 2008; Hamilton et al., 2018) as LCE destruction/formation
involves specific processes (Frolov et al., 2004; Donohue et al., 2016). We therefore focus on the LCEs
contained in the central part of the GoM from 86°W to 94°W. Annual composites are computed along
with monthly composite averages in order to assess seasonal variability. Composite LCEs averaged
during the months of January and February are referred to as winter composites and those averaged
during July and August are referred to as summer composites. These composites provide an overview
of the LCEs mean hydrographical, biogeochemical and dynamical characteristics.
**II.4/ Diagnostics**

The LCE radius $R_{LCE}$ is estimated as the radial distance between the center and the peak

azimuthal velocity $V_{max}$. The mixed layer depth (MLD), a major physical factor influencing nutrient
distribution and [CHL] dynamics (Mann and Lazier, 2006), is defined as the depth at which potential
density exceeds its value at 10m depth by 0.125 kg·m$^{-3}$ (Levitus, 1982; Monterey and Levitus, 1997).
An important driver of the mixed layer deepening is the stratification of the water column, which is
evaluated by the square of the buoyancy frequency $N^2(z) = \frac{-g}{\rho_0}\frac{\partial \rho}{\partial z}$, where g is the gravitational
acceleration, z is depth, $\rho$ is density and $\rho_0$ is a reference density.

As carried out in Damien et al. (2018), several metrics are defined and used to describe [CHL]:





• [CHL]$_{surf}$: [CHL] averaged between 0 and 30 m depth, and considered as surface concentration

(in mg CHL·m$^{-3}$),

• [CHL]$_{tot}$: integrated content of [CHL] over the 0-350 m layer (in mg CHL·m$^{-2}$),
• DCM: depth of the Deep Chlorophyll maximum (in m),
• [CHL]$_{DCM}$: [CHL] value at DCM depth (in mg CHL·m$^{-3}$).
To understand the mesoscale distribution of [CHL], key biological variables are vertically integrated
between 0 and 350m: the phytoplanktonic concentration [PHY]$_{tot}$, the primary production rate PP$_{tot}$ and
the grazing rate GRZ$_{tot}$. PP$_{tot}$ consists of two components: new production PPN$_{tot}$ fueled by nutrients
supplied from a source external to the mixed layer and regenerated production PPR$_{tot}$ sustained by
recycled nutrients within the euphotic layer (Dugdale & Goering, 1967; Eppley & Peterson, 1979). A
normalized chlorophyll concentration anomaly within LCEs, [CHL]', is also computed as
$$[CHL]' = \frac{[CHL] - \overline{[CHL]}}{SD([CHL] - \overline{[CHL]})}$$    , where $\overline{[CHL]}$ is the averaged background [CHL] field in the open
GoM waters (for radius>250km from the LCEs' centers) and SD the standard deviation operator,
following a similar approach as Gaube et al. (2013, 2014) and Dufois et al. (2016). To limit the
influence of very high [CHL] values in coastal waters under the direct influence of continental
discharges, a salinity filtering criterion (lower than 36 psu) is applied. A similar method was used by
Gaube et al. (2013, 2014) to filter edge effects but using a distance criterion instead.
**III/ Results**

**III.1/ Satellite observations of [CHL]**



Fig 1 shows the 8-day averaged satellite observations of the surface chlorophyll around May 29[th]
2003 (a) and October 19[th] 2004 (b). These observations highlight the strong contrast between the
eutrophic conditions in the coastal waters and the oligotrophic conditions in the open ocean, as already
addressed by several studies (Martinez-Lopez & Zavala-Hidalgo, 2009; Pasqueron de Fommervault et
al., 2017). Far from the coast, these figures also reveal that the surface chlorophyll varies at a scale of
the order of 100km with a distribution that tends to follow the absolute dynamic topography (ADT)
contours.

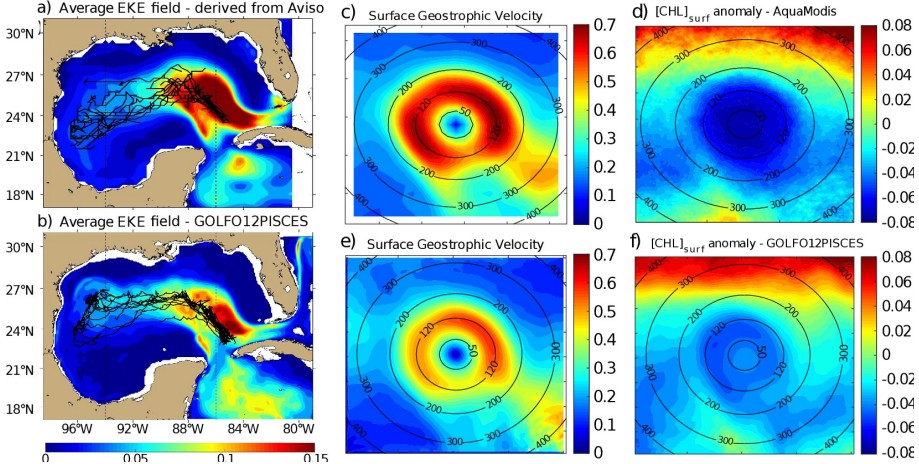

**Figure 2: Average eddy kinetic energy (EKE) field derived from (a) Aviso geostrophic surface velocities and from (b) GOLFO12-**
**PISCES currents at 10m depth. The trajectories of the tracked LCEs are superimposed to the EKE field (black lines). Vertical**
**black dashed lines indicate the central GoM area over which composites are built. Annual LCE composite images of surface**
**geostrophic velocities for (c) Aviso images and (e) GOLFO12-PISCES. Annual LCE composite images of surface chlorophyll**
**concentration anomaly for (d) Modis images and (f) GOLFO12-PISCES. Black circles indicate the radius in kilometers.**
LCEs trajectories are reported on Fig 2.a, superimposed onto the geostrophic climatological eddy
kinetic energy (EKE) field at the surface. EKE is computed from eddy velocities defined on each grid
cell as the difference between the total horizontal current and its mean value over 120 days. This time





window is chosen to filter the seasonal signal. EKE is concentrated in the LC and on the westward
pathway of the LCEs (Lipphardt et al. 2008) demonstrating that LCEs constitute the major source of
EKE in the GoM open waters (Sheinbaum et al., 2016; Sturges & Leben, 2000; Hamilton, 2007;
Jouanno et al., 2016).

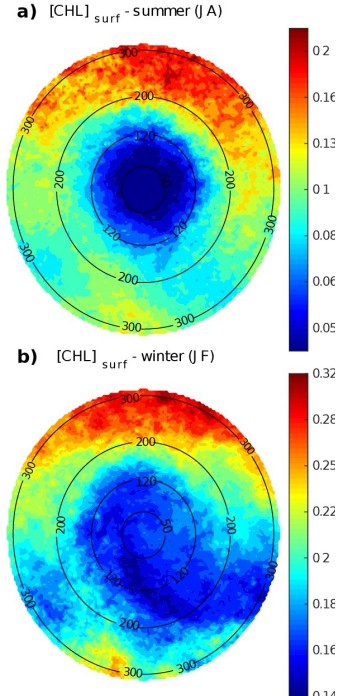

**Figure 3: LCE composite images of [CHL]$_{surf}$ derived from Aqua-MODIS for the (a) summer and (b) winter seasons. Black circles**
**indicate the radius in kilometers.**
LCE annual composites of surface geostrophic velocities (Fig 2.c) and [CHL]$_{surf}$ (Fig 2.d) are
built from 482 different satellite images. On average, we found that R$_{LCE}$ ~ 120 km and V$_{max}$ ~ 0.6-0.7
m·s$^{-1}$, in agreement with previously reported LCEs (Elliot, 1982; Cooper et al., 1990; Forristal et al.,
1992; Glenn and Ebbesmeyer, 1993; Weisberg and Liu, 2017; Tenreiro et al., 2018). LCEs are
associated with a negative [CHL]$_{surf}$ anomaly (~ -0.07 mg.m$^{-3}$ in the annual average). The LCEs



influence on [CHL]$_{surf}$ is largest in summer (Fig 3.a) when it reaches very low values (< 0.045 mg·m$^{-3}$),
which corresponds to an anomaly of ~ -0.08 mg·m$^{-3}$. This anomaly is less marked in winter (~ -0.06
mg.m$^{-3}$, Fig 3.b) when [CHL]$_{surf}$ ~ 0.17 mg·m$^{-3}$ within LCEs. The high chlorophyll concentrations in the
northern part of the composites (in the southern part too but in smaller proportions) are related to
shelves.
**III.2/ Dynamical characterization of modeled LCEs**
A total of 11 model LCEs were detected during the 5 years of simulation. Their trajectories are
reported in Fig 2.b, superimposed upon the climatological EKE field simulated at 10 meters. The
westward / southwestward propagation of LCEs is well reproduced (Vukovich, 2007) even though the
LCEs translation is almost zonal in GOLFO12-PISCES. Comparison with Fig 2.a shows the ability of
GOLFO12-PISCES to represent the mean and transient dynamical features of the GoM open waters
(see also Garcia-Jove et al., 2016).
The robustness of the composite method arises from the number of LCE images used to build the
composites:
•     Annual composite is built from 605 5-day averaged LCE pictures from 10 different LCEs,
•     Summer composite is built from 83 5-day averaged LCE pictures from 8 different LCEs,
•     Winter composite is built from 93 5-day averaged LCE pictures from 9 different LCEs.
The                                                                                              with
velocities                                                                                  the surface
signature                                                                                       tal

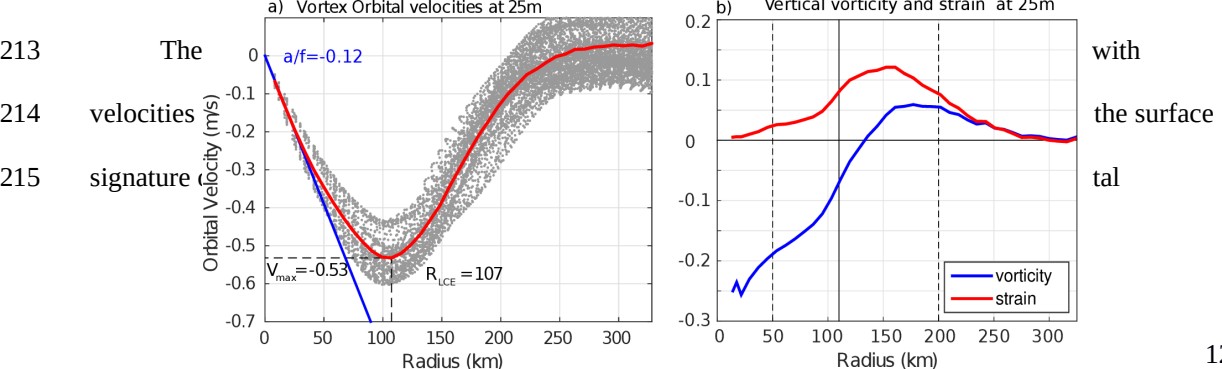





velocities (~ 25% on average over the 50-200 km radius range). This bias could result from the
relatively coarse model resolution and 5-day output frequency that are unable to fully capture the
gradient intensity at $R_{LCE}$. The assumption of an axial symmetry of the LCE circulation around its
center also induces an error that tends to decrease $V_{max}$.
**Figure 4: (a) Orbital velocities at 25m depth in function of the radius of each detected LCE (light gray dots). The red line is the**
**LCE orbital velocity profile of the annually-averaged composite. (b) Vertical vorticity and strain computed from the averaged**
**orbital velocity profile assuming no radial velocity in cylindrical coordinates as**   $\zeta_z = \dfrac{1}{f\,r}\dfrac{\partial\,rv}{\partial\,r}$   **and**   $S = \dfrac{1}{f}\left(\dfrac{\partial\,v}{\partial\,r} - \dfrac{v}{r}\right)$ .

Orbital velocities of composite eddies are used to distinguish different dynamical areas within

LCEs. The model annual average dynamical profile at 25m depth (Fig 4) reveals a typical vortex-like
structure with $R_{LCE}$ ~ 107 km and $V_{max}$ ~ 0.53 m·s$^{-1}$ and suggests the following decomposition:
• r < 50 km : the **LCEs core**, where the eddy is approximately in solid body rotation: $V_{orb}$ = a·r

where the coefficient a is related to the Rossby number (Ro = 2a/f ). The ratio a/f is estimated

to be ~ -0.12 (Fig. 4). In this field, the stain is reduced to a minimum and the flow is dominated

by rotation.

• 50 km < r < 200 km: the **LCEs ring** structure where the orbital velocity reaches its maximum

at $R_{LCE}$ and then decreases. The horizontal strain is important in this field, even dominating

vorticity from radius exceeding  $R_{LCE}$.

•  R > 200 km: the **background GoM**, where the velocity anomalies related to the LCE vanish.

In the vertical (Fig 5.a), LCEs are near-surface intensified anticyclonic vortex rings. At depth,

the orbital peak velocity decreases rapidly. At 500 m depth, $V_{max}$ ~ 0.17 m·s$^{-1}$ and $R_{LCE}$ ~ 75 km, and



the dynamical LCE signal nearly vanishes below 1500 m depth ($V_{max} < 0.03$ m·s$^{-1}$). The proposed
division into 3 distinct dynamical regions applies from the surface down to 500 m depth (Fig 5.a).

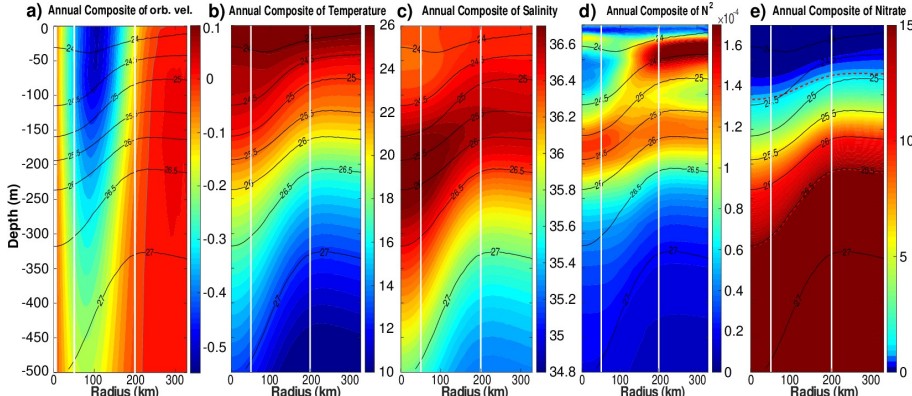

**Figure 5: Annually-averaged LCE composite transects of (a) orbital velocities [m/s], (b) potential temperature [°C], (c) salinity [psu], (d) squared Brunt-Väisälä frequency (N$^2$ in s$^{-2}$) and (e) nitrate concentration [mmol·m$^{-3}$]. Isopycnals anomalies (black contours) are superimposed on all panels. Vertical white lines delimit the three dynamical fields of the LCE composite. On panel e, dashed red lines highlights two specific iso-nitrate contours: 1 and 15 mmol·m$^{-3}$.**

The composite hydrological structure of modeled LCEs is shown in Fig 5.b and 5.c. The

depression of isopycnals, associated with a depression of isotherms and isohalines, is characteristic of
oceanic anticyclones. In the core of the eddies, the composite depicts a salinity maximum located
between 100 and 300 m, corresponding to the signature of the Atlantic Subtropical UnderWater
(ASTUW) of Caribbean origin entering the GoM through the Yucatan Channel (Badan et al., 2005;
Hernandez-Guerra & Joyce, 2000; Wuust, 1964). This salinity maximum is not limited to the core of
the LCE but gradually erodes and shallows: 36.82 psu at 200 m in the LCEs core and 36.61 psu at 150
m in the background GoM common water. Details on the fate of this salinity maximum investigated
with GOLFO12 simulations can be found in Sosa-Gutiérrez et al. (2020). The ASTUW layer (salinity >
36.5 psu) is also thicker in the LCEs core (~190 m thick) compared to the background GoM water





(~120 m thick). Overall, GOLFO12-PISCES reproduces the observed hydrological structure of LCEs
(Elliott, 1982; LeHenaff et al., 2012; Hamilton et al., 2018; Meunier et al., 2018b).

The annually averaged LCE composite presents a lens-shaped structure exhibiting a ~50 m thick

layer of weakly stratified waters located between 50 and 100 m depth (Fig 5.d). This subsurface modal
water presents hydrological characteristics close to the observed background GoM waters (potential
temperature ~25.4°C and salinity ~ 36.3 psu, Meunier et al., 2018b) and is surrounded below and above
by well stratified layers (Meunier et al., 2018a). The upper pycnocline varies seasonally and vanishes in
winter due to the deepening of the mixed layer, whereas the lower pycnocline is permanent.

The downward displacement of isopycnals is associated with a depletion of nutrients in the upper

layer of the LCEs core (Fig 5.e). This is a typical feature of mesoscale anticyclones in the ocean
(McGillicuddy et al. 1998; Oschlies and Garcon, 1998). The 1 mmol.m$^{-3}$ iso-nitrate concentration
(hereafter $Z_{NO3}$, sometimes referred to as the nitracline as in Cullen & Eppley, 1981; Pasqueron de
Fommervault et al., 2017 or Damien et al., 2018) is located at ~ 70 m depth in the background GoM
waters whereas it is found much deeper in the core ($Z_{NO3}$ ~ 106 m). At depth, iso-nitrate layers and
isopycnals are well correlated (Ascani et al., 2013; Omand & Mahadevan, 2014). For instance, iso-
nitrate concentration of 15 mmol·m$^{-3}$ follows the displacements of the 1026.5 kg·m$^{-3}$ isopycnal.
However, above 150 m, the density/nitrate relation is different inside and outside the eddies ($Z_{NO3}$ is
collocated with isopycnal 1024.4 kg·m$^{-3}$ in the LCEs core while it is on isopycnal 1024.9 kg·m$^{-3}$ in the
background GoM).

**III.3/ Surface and vertical distribution of chlorophyll in LCEs**



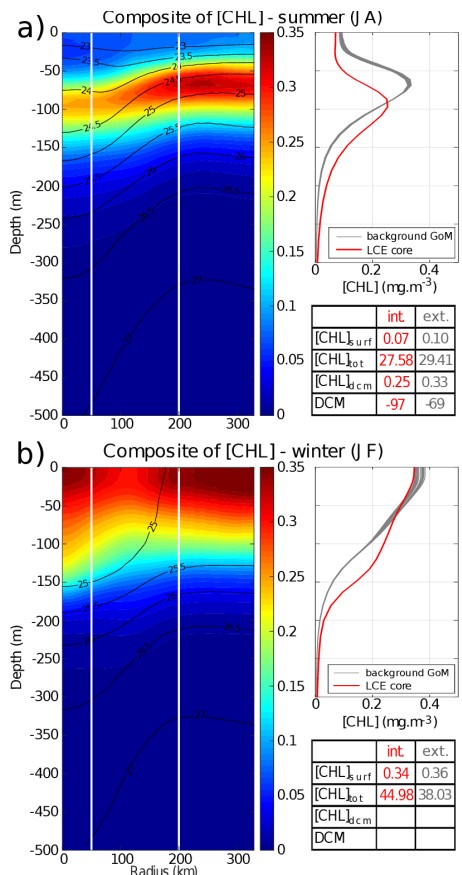

**Figure 6: LCE composite transects of [CHL] during summer season (A) and winter season (B). Density anomalies (black**

**contours) are superimposed. Vertical white lines delimit the three dynamical fields of the LCE composite. For each season, [CHL]**

**profiles in the LCE core (r < 50 km, red lines) and in the background GoM (200 km < r < 330 km, gray lines) are plotted. Key**

**metrics concerning [CHL] profiles are also indicated in the tables.**

The large difference in stratification between the LCEs core and background GoM suggests a

contrasted seasonal response of the [CHL]. This is confirmed by the analysis of summer and winter

composites of [CHL] vertical distribution:

- In summer (Fig 6.a), $[CHL]_{surf}$ is ~ 30% lower in the LCEs core (r < 50km) than in the

    background GoM (200 km < r < 330 km). A pronounced DCM, characteristic of oligotrophic

    environments, is deeper in the core (~ 97 m) than in the background GoM (~ 69 m) with

    chlorophyll concentrations significantly lower in the interior (~ - 25%).

• In winter, the [CHL] is maximum at the surface in all the composite domains (Fig 6.b).
[CHL]$_{surf}$ is lower in the LCEs core compared to the background GoM but the difference is less
marked (~ - 6%) than in summer. The main discrepancy is the depth of the inflection point of
these profiles. It is deeper in the LCEs core (~-150 m), resulting in a more homogenized [CHL]
over a deeper layer than in the background GoM (~-120 m).
However, despite reduced surface concentration both in winter and summer, the integrated
chlorophyll content, [CHL]$_{tot}$, shows a distinct seasonal pattern compared to the surface (tables in Fig

6):

• In summer, [CHL]$_{tot}$ is lower in the LCEs core (27.58 mg·m$^{-2}$) compared to the background
GoM (29.41 mg·m$^{-2}$) and $\Delta$[CHL]$_{tot}$ = -1.83 mg·m$^{-2}$,
• In winter, [CHL]$_{tot}$ is higher in the LCEs core (44.98 mg·m$^{-2}$) compared to the background GoM
(38.03 mg·m$^{-2}$) and $\Delta$[CHL]$_{tot}$ = + 6.95 mg·m$^{-2}$.
The winter increase of [CHL]$_{tot}$ is around 29% in the background GoM whereas it reaches 63% in the
LCEs core, leading to [CHL]$_{tot}$ in the core being larger than [CHL]$_{tot}$ in the background GoM in winter.
Meanwhile, [CHL]$_{surf}$ remains lower within the LCEs core. The fact that the [CHL] at the surface does
not reflect its depth-integrated behavior means that the peculiar variability of [CHL] within LCEs may
not be fully captured by ocean color satellite measurements. This is consistent with Pasqueron de
Fommervault et al. (2017) and Damien et al. (2018) observations and modeling results which addressed
the vertical [CHL] distribution in the GoM.



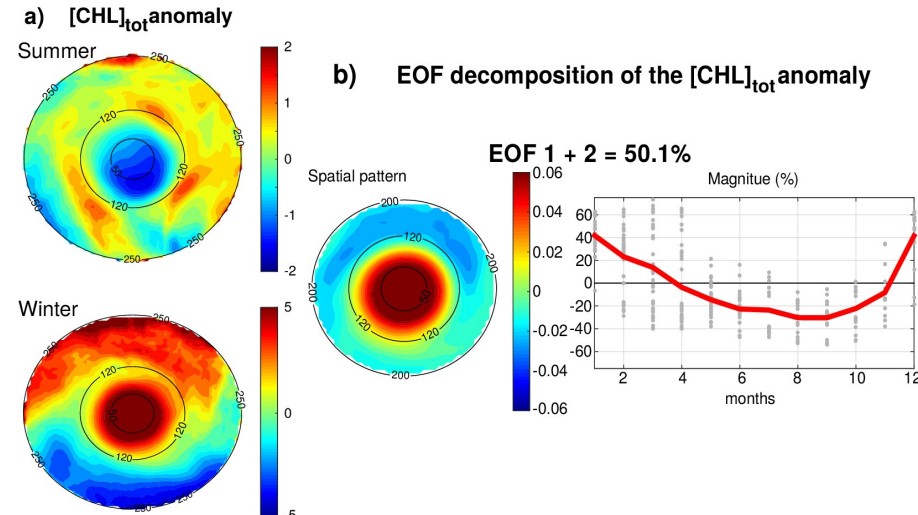

**Figure 7: (a) Anomaly of [CHL]$_{tot}$ in summer and winter seasons. Black circles indicate the radius in kilometers. (b) EOF**
**decomposition of the [CHL]$_{tot}$ anomaly. The spatial patterns and monthly magnitude (gray dots; the red line represents their**
**monthly averaged value) of the two first modes are indicated. Modes 1 and 2 were summed together (upper panel) and represent**
**50.1% of the total variance.**

[CHL]$_{tot}$ is strongly shaped by both the seasonal variability and the LCEs. The seasonal

composites of [CHL]$_{tot}$, shown in Fig 7.a, confirm the summer/winter contrast and highlight a
monopole structure with a relatively homogeneous distribution of [CHL]$_{tot}$ within the eddy's core. In
order to better characterize the spatio-temporal variability of [CHL]$_{tot}$ induced by LCEs, an Empirical
Orthogonal Function (EOF) analysis was performed on the [CHL]$_{tot}$ anomaly (Fig 7.b) following the
methodology of  Dufois et al. (2016). It consists in decomposing the signal into orthogonal modes of
variability. Here, we have chosen to focus on the first two most significant modes which explain 40.2%
and 9.9% of the variability. Since they both depict a similar monopole structure in the LCEs core, they
were added up in a mode referred to EOF 1+2 responsible for 50% of the total [CHL]$_{tot}$ variance within
LCEs. The third eigenmode (not shown) accounts for 6.2% and depicts a dipole structure with opposite
polarity located at the east and north of the eddy center. On average, the EOF1+2 mode is positive in



winter (from December to March) and negative the rest of the year (from April to November), with a
maximum in January December and a minimum in September. This justifies, a posteriori, the choice to
consider winter and summer LCE composites.

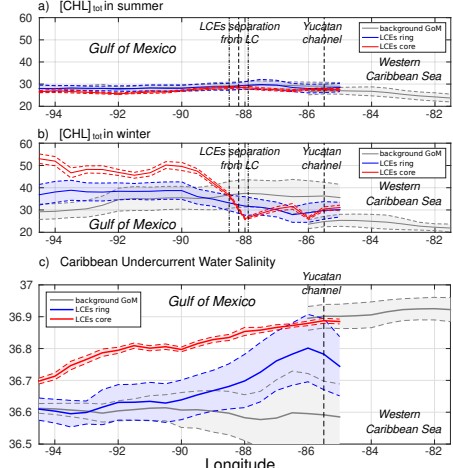

**Figure 8: (a) Summer [CHL]$_{tot}$, (b) winter [CHL]$_{tot}$ and (c) salinity of Caribbean waters (ASTUW defined as the subsurface**
**salinity maximum) as a function of longitude in (red) the LCEs core, (blue) the LCEs ring and in (gray) the background GoM.**
**Full lines indicate the averaged value and dashed lines the +/- one standard deviation interval.**

The composite evolution of the LCEs [CHL]$_{tot}$ along their westward journey is shown in Fig 8.a

and 8.b. It illustrates how the total chlorophyll concentration is preferentially increased in winter within
the LCEs core, as soon as the LCEs are shed from the LC. The winter [CHL]$_{tot}$ within LCEs is much
larger (exceeding one standard deviation) than the background winter [CHL]$_{tot}$. In terms of integrated
[CHL], the LCEs-induced seasonal variability overwhelms the GoM open-waters background seasonal
variability.
**IV/ Discussion**



In an oligotrophic environment such as the GoM open-waters, the primary production is
generally limited by nutrient supply and [CHL]$_{tot}$ exhibits low seasonal variability at the GoM basin
scale (Pasqueron de Fommervault et al., 2017). The winter increase of [CHL]$_{tot}$ within the LCEs core
(which translates into an effective increase of biomass, see appendix A) contrasts and may have large
implications for the regional biogeochemical cycles and ecosystem structuration. It also echoes several
studies which report elevated [CHL]$_{surf}$ within anticyclonic eddies in the oligotrophic subtropical gyre
of the southeastern Indian Ocean (Martin and Richards, 2001; Waite et al., 2007; Gaube et al., 2013;
Dufois et al., 2016, 2017; He et al., 2017), questioning the classical paradigm of low productivity
usually associated with anticyclonic eddies.
The mechanisms explaining the LCE impact on [CHL] are discussed below, trying to rationalize
the respective role of abiotic (e.g., trapping, winter mixing, Ekman pumping) and biotic processes (e.g.,
primary production (PP), grazing pressure, regenerated versus new PP).
**IV.1 Eddy trapping**
The distinct hydrological and biogeochemical properties associated with the LCEs core suggest
their ability to trap and transport oceanic properties. This mechanism, known as the eddy-trapping
(Early et al., 2011; Lehahn et al., 2011; McGillicuddy, 2015; Gaube et al., 2017) is efficient only if the
orbital velocities of the vortex are faster than the eddy propagation speed (Flierl, 1981; d'Ovidio et al.,
2013). The rotational velocities of the model LCEs are ~ 0.53m·s$^{-1}$ are one order of magnitude larger
than the propagation velocities (~ 0.046 m·s$^{-1}$ on average). This suggests that LCEs might have a



certain ability to trap the water masses present in their core with relatively low exchanges with the
exterior.

Salinity is well-suited to investigate water masses trapped within the LCEs core during their

propagation toward the western GoM (Fig 8.c; Sosa-Gutierez et al., 2020): salinity distribution shows a
marked subsurface maximum that it is not affected by biogeochemical processes. In the Western
Caribbean Sea, ASTUW is characterized by high salinity (~ 36.9 psu on average) and low standard
deviation (< 0.05 psu). The eastern GoM salinity field reveals that most of the ASTUW crosses the
Yucatan Channel within the Loop Current. During the formation of LCEs, a significant part of
ASTUW is captured into the LCEs core with low alteration of its properties (Fig 5.c and 8.c). Within
the LCEs core, the water mass is transported from eastern to the western GoM where its salinity
decreases from 36.9 psu to 36.7 psu. Although altered, the ASTUW signature is still clearly detectable
in the GoM western boundary. The other part of ASTUW entering the GoM is found in the LCEs ring.
Compared to the core, the salinity in the ring is on average lower (~ 36.8 psu in the eastern GoM) and
presents a high standard deviation, pointing out that more recent ASTUW co-exists with older ASTUW
that yields eroded salinity maxima. As LCEs travel westward across the GoM, salinity in the LCEs ring
decays rapidly to reach values similar to the background GoM values (~ 36.6 psu). This
homogenization mainly arises from vertical mixing and winter mixed layer convection (Sosa-Gutierez
et al., 2020). Horizontal intrusions and filamentation may also contribute to this homogenization
(Meunier et al., 2020). The composites also suggest that almost no ASTUW enters the GoM apart from
the LCEs. The slight increase of the background salinity from eastern to western GoM is a consequence
of the diffusion of salt from the LCEs toward the exterior.

Although LCEs undergo considerable decaying rates, their erosion is particularly strong in the

ring while the core remains better isolated from the surrounding waters (Lehahn et al., 2011; Bracco et



al., 2017). Given that the LCEs core is also quite homogeneous, the following discussion relies on the
analysis of the seasonal cycles of selected parameters averaged within the LCEs core.
**IV.2 Nitracline depth and nutrient supply into the mixed layer**

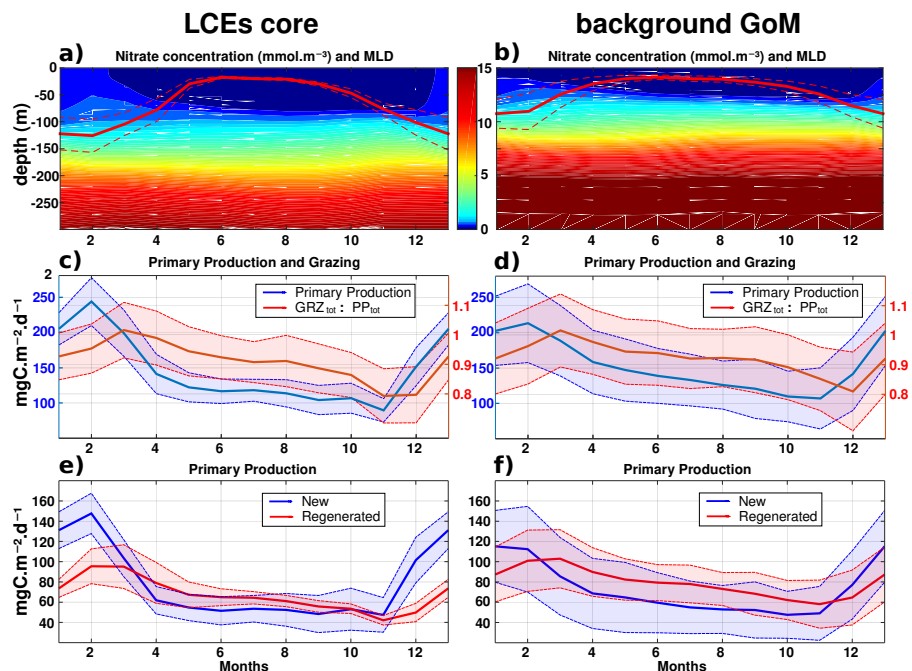

**Figure 9: Climatological seasonal cycles of (a and b) nitrate concentration profiles (the red line overlaid is the average mixed layer**
**depth), (c and d) the total primary production (blue) and the ratio of grazing rate over primary production (red) and (e and f) the**
**new (blue) and regenerated (red) primary production. The left panels (a, c and e) refer to the seasonal time series in the LCEs**
**core (r < 50 km) whereas the right panels (b, d and f) refer to the seasonal time series in the background GoM (r > 200 km). For**
**each average cycle, the mean value is shown (full line) along with its variability (+/- 1 standard deviation relative to the mean,**
**dashed lines).**
The LCEs impact the upper ocean stratification (Fig 5.d), the nutricline depth (Fig 5.e) and
consequently the nutrient supply to the euphotic layer (McGillicuddy et al., 2015). The relationship





between mixed layer deepening and nutrient supply is studied here by comparing the $Z_{NO3}$ with the
MLD (Fig 9.a,b).

In late-spring and summer (from May to September), the water column is stratified (shallow

MLD) and the downward displacement of the isopycnals within the LCEs pushes nutrients below the
euphotic zone (see also Figs 5.e, 6.a): less nutrients are available within the LCE cores for
phytoplankton growth, explaining a deeper and less intense DCM. In winter, the convective mixing,
fostered both by intense buoyancy losses and strong mechanical energy input at the surface, causes a
larger deepening of the mixed layer within the LCEs core (~ - 125 m, Fig 9.a) compared to the
background (~ - 85 m, Fig 9.b). This asymmetry is due to a pronounced decrease of the surface and
subsurface stratification within the LCE core (Fig 5.d, Kouketsu et al., 2012). A quantitative diagnostic
of the stratification is given by the columnar buoyancy, $\int_{0}^{H} N^2(z) . z . dz$ which measures the buoyancy
loss required to mix the water column to a depth H (Herrmann et al. 2008). Fig 10.a reveals significant
differences in pre-winter buoyancy between the eddy core and its surroundings. Assuming that the
change in buoyancy content is mainly controlled by the buoyancy flux at the surface (see Turner 1973;
Lascaratos & Nittis, 1998), it suggests that mixing the water column down to ~ -210 m depth requires
smaller surface buoyancy loss in LCEs cores compared to the background GoM (Fig 10.b).

However, the larger winter deepening of the mixed layer within the LCEs core is not a sufficient

condition to explain a larger nutrient supply. Indeed, it fosters the transport of nutrients from the
nitracline toward the mixed layer because both are getting closer. Fig 10.c highlights that a smaller
buoyancy loss mixes down the water column to greater nutrient concentration levels in the LCEs core
compared to the LCEs surrounding. This likely explains the winter increase of surface nitrate



concentration within the LCEs (Fig 9.a). In addition, a diagnostic of the different contributions to
[NO₃] evolution is proposed in appendix B. It shows the dominant role of vertical advection and
diffusion in winter in providing nutrients to the euphotic layer in the LCEs core.
**Figure 10: (a) Columnar Buoyancy transect composite in summer. Iso-nitrate concentrations (black contours) are superimposed.**

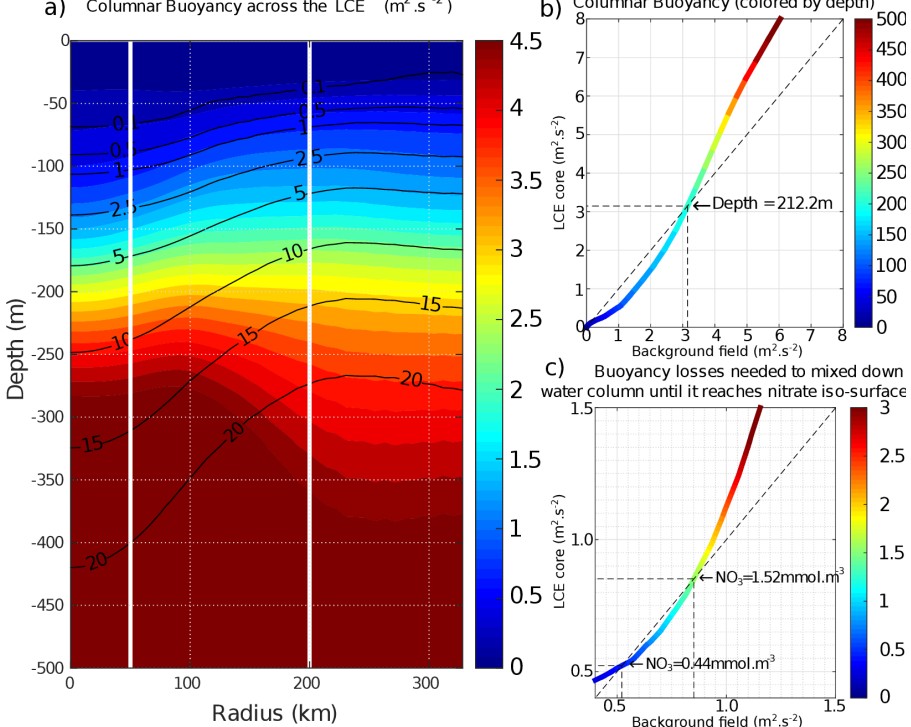

**Vertical white lines delimit the three dynamical fields of the LCE composite. (b) Vertical increase of the columnar buoyancy in**
**the LCEs core versus the background GoM. Colors refer to depth. (c) Columnar buoyancy loss required to mix the water column**
**down to the iso-nitrate surface defined by the line color.**

So far we have assumed that the surface buoyancy fluxes are identical over the LCEs core and

the background GoM. However, this is not strictly the case because temperature/salinity features in the
LCEs and background waters are different (Fig 5.b,c; see also Williams 1988). The modeled surface
buoyancy loss during winter season is ~18 % more intense within the LCEs. This difference is
substantial and probably mainly driven by additional surface cooling applied on the warm LCE core





through air-sea interaction. It contributes to enhance convection within the eddies core, and then
nutrient supply toward the surface.

### IV.3 Productivity

The primary productivity $PP_{tot}$ presents a clear seasonal cycle both in the LCEs cores and in the
background GoM with lower values in October-November, a sharp increase starting in November, a
maximum in February and a gradual decrease from March to October (Fig 9.c and 9.d). The pressure
exerted by zooplankton grazers varies seasonally. It shows a similar seasonal cycle in the LCEs core
and in the background GoM. On average, ~ 90% of the total daily growth is consumed by grazing,
reaching the highest impact in March, just one month after the peak season of the $PP_{tot}$ in both LCEs
dynamical areas. The annual $PP_{tot}$ is slightly lower in the LCEs core (~ 142.4 mgC·m$^{-2}$.d$^{-1}$) than in the
background GoM (~ 148.9 mgC·m$^{-2}$.d$^{-1}$). The amplitude of the seasonal cycle is larger in the LCEs
core: from April to November, $PP_{tot}$ is on average ~12% lower in the LCEs core whereas, in winter,
$PP_{tot}$ is ~14% higher where it reaches ~ 243.2 mgC·m$^{-2}$.d$^{-1}$ in February.
The ratio of the $PPN_{tot}$ and $PPR_{tot}$ provides information about the mechanisms controlling the
biomass growth (Fig 9.e and 9.f). In winter, the $PPN_{tot}$ plays a leading role, reaching up to 113-147
mgC·m$^{-2}$·d$^{-1}$, driven by the winter mixing and induced $NO_3$ fluxes (see Appendix B). Conversely, the
$PPR_{tot}$ is dominant from April to October. During this period, low $NO_3$ resources are available in the
euphotic layer and the ecosystem preferentially uses ammonium to sustain the $PP_{tot}$. This seasonal
pattern is characteristic of oligotrophic environments such as the GoM open waters (Wawrik et al.,
2004; Linacre et al., 2015).



In winter, changes in $PP_{tot}$ are correlated to the intensity of winter mixing in the LCEs core (Fig
9.c) and the background GoM (Fig 9.d). The larger $PPN_{tot}$ in the eddy core is consistent with a larger
supply of $NO_3$ and evidences that the core of anticyclones can be preferential spots of enhanced
biological production.
**IV.4 How to explain summer productivity?**
In summer, the total primary production is higher in the background GoM waters as the
regenerated production rate is higher. But surprisingly, the new primary production exhibits similar
rates in both regions, although $NO_3$ depletion occurs deeper in the LCEs core. In the absence of a
strong enough vertical mixing when the mixed layer is shallow, this apparent mismatch requires an
additional mechanism, vertical advection, capable to supply $NO_3$ to the euphotic layer (Sweeney et al.,
2003; McGillicuddy et al., 2015).
The model vertical velocity in the LCEs reveals an upward pumping in their core (Fig 11). The
vertical velocity between 100 and 500 m is on average + 0.07 m·day$^{-1}$. This vertical transport is mainly
driven by two mechanisms, eddy pumping (Falkowski et al., 1991) and eddy-wind interaction (Dewar
and Flierl, 1987), but their relative importance is difficult to quantify (Gaube et al. 2014; McGillicuddy
et al., 2015).
The eddy pumping mechanism is related to the decay of the rotational velocities from the
moment LCEs are released from the Loop Current. In the LCE core, this decay is considered as
moderate since lateral diffusivity is expected to be relatively low (section V.1). This process may
however be considerable in the LCE ring where the erosion rates are important (Meunier at al., 2020).



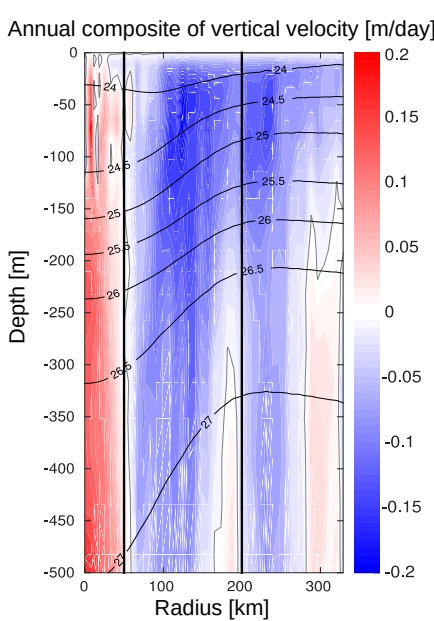

**Figure 11: Annually-averaged LCE composite transects of vertical velocities (m/day). Isopycnals anomalies (black contours) are**
**superimposed on all panels. Vertical white lines delimit the three dynamical fields of the LCE composite.**
Eddy-wind interactions are due to mesoscale modulation of the Ekman transport. Following the
observation of a LCE core in quasi-solid body rotation, the horizontal vorticity varies little with the
radius resulting in a negligible "non-linear" contribution of the Ekman pumping (McGillicuddy et al.,
2008; Gaube et al., 2015). Assuming a small effect of the eddy SST-induced Ekman pumping, the total
Ekman pumping simplifies into its "linear" contribution computed as $W_E = \dfrac{\nabla \times \tau}{\rho_0 \cdot (f + \zeta)}$, where $\rho_0$ is the
surface density, f the Coriolis parameter, τ the stress at the sea surface depending on both the wind and
ocean currents at the surface (Martin and Richards, 2001, equation 12) and $\nabla \times$ the curl operator.
Considering uniform wind velocities ranging from 4.5 to 7.5 m·s$^{-1}$ (Nowlin & Parker, 1974;
Passalacqua et al., 2016) blowing over the LCE, the curl of the stress arises from the anticyclonic





surface circulation generated by the eddy. Its manifestation is a persistent horizontal divergence at
surface balanced by an upward pumping in the eddy interior (see Martin & Richards, 2001; Gaube et
al., 2013, 2014 for further details). With $\rho_0 \sim 1023$ kg·m$^{-3}$ and $f \sim 6.2.10^{-5}$ s$^{-1}$, we estimate $W_E$ to be in
the order of $+ 0.06$-$0.13$ m·day$^{-1}$, in agreement with the modeled vertical velocity within the core. The
Ekman-eddy pumping mechanism could explain a large fraction of the gradual upwelling of isopycnals
within the eddy's core and may actively contribute to the advective vertical flux of nutrients (see
Appendix B). In summer, this mechanism could explain why new primary production rates are similar
in the LCEs core and the background GoM waters although the nutrient pool is located much deeper in
the LCEs core.

The eddy-Ekman pumping persists in the LCEs core throughout their lifetime as long as there is

a wind stress applied at the surface. During wintertime, we expect that both vertical mixing and eddy-
Ekman pumping participate to increase the new primary production. A question then arises on the
relative contribution of winter mixing to eddy-Ekman pumping in the LCEs core primary production
increase in winter. This issue was tackled by He et al. (2017) and Travis et al. (2019) comparing the
rate of change of the mixed layer depth with the vertical velocity induced by the eddy-Ekman pumping
(equation 4 in He et al, 2017). In the LCEs core, we estimate the mixed layer to deepen at roughly 0.8
m·day$^{-1}$, which is on average about 10 times larger than pumping mechanism. This supports winter
mixing as the overwhelming process for the LCEs-induced primary production peak in winter.
**V/ Summary and perspectives**

The [CHL] variability induced by the mesoscale Loop Current Eddies in the Gulf of Mexico is

studied by analyzing vortex composite fields generated from a coupled physical-biogeochemical model



at 1/12° horizontal resolution. LCEs are hotspots for mesoscale biogeochemical variability. Despite the
$[CHL]_{surf}$ negative anomaly associated with their core (r < 50 km), model results indicate that LCEs are
associated with enhanced phytoplankton biomass content, particularly in winter. This enhancement
results from the contribution of multiple mechanisms of physical-biogeochemical interactions and
contrasts with the background oligotrophic surface waters of the GoM.

The main results of this study are:

•   LCEs cores present a negative surface chlorophyll anomaly,
•   Unlike $[CHL]_{surf}$, $[CHL]_{tot}$ is larger in the LCEs cores compared to the background GoM in

winter.

•   LCEs core trigger a large phytoplankton biomass increase in winter,
•   The winter mixing is a key mesoscale mechanism that preferentially supplies nutrients to the

euphotic layer within the LCEs core. Consequently, it drives an eddy-induced peak of new

primary production,

•   Ekman-eddy pumping is a significant mechanism for sustaining relatively high new primary

production rates within LCE cores during summer.

The phytoplankton biomass increase in individual LCEs cores suggests that LCEs play an important
role in sustaining the large-scale GoM productivity.

Although the biological response to LCEs may present some specificities due to the particular

dynamical nature of LCEs, this study suggests potentially generic insights on the biogeochemical role
that anticyclonic eddies could play in oligotrophic environments. It echoes the previous works of
Martin and Richards (2001), Gaube et al. (2014, 2015) and especially Dufois et al. (2014, 2016) and He
et al. (2017) who proposed winter vertical mixing as an explanation for the positive $[CHL]_{surf}$ anomaly
observed in anticyclones in the South Indian Ocean. One of the most crucial points to be underlined





from our results is that the enhanced primary production and biomass content within anticyclonic
eddies may not necessarily be correlated with the surface layer variability. In oligotrophic areas, the
integrated content of chlorophyll in the water column has to be considered. This implies that caution
should be exercised in the analysis and interpretation of $[CHL]_{surf}$ observed by remote sensing
instruments and highlights the crucial need for in-situ biogeochemical and bio-optical measurements.
In oligotrophic environments, defined by their low production rates and their low chlorophyll
concentration, anticyclonic eddies are able to trigger local enhanced biological productivity and
generate phytoplankton biomass positive anomalies. In a scenario of expansion of oligotrophic areas
(Barnett et al., 2001; Behrenfeld et al., 2006; Polovina et al., 2008), the fate and role of mesoscale
anticyclones is an important aspect to be considered.
This study focuses on mesoscale physical-biogeochemical interactions which is the spectral
range resolved by GOLFO12-PISCES configuration. To go further into the analysis of anticyclonic
eddies in oligotrophic environments, the role of submesoscale is of particular interest since it has been
proved to trigger mechanisms of significance importance for biogeochemistry (Levy et al., 2018).
Higher model resolutions can locally enhanced density gradients (Levy et al., 2012; Omand et al.,
2015) leading to ageostrophic circulations that perturbs the circular flow around vortices (Martin and
Richards, 2001) or enhanced vertical velocities that potentially foster the nutrient supply to the
euphotic layer. Beside the mesoscale Ekman pumping located at the eddy center, eddy-wind
interactions also produce vertical velocities at the eddy periphery (e.g. Flierl and McGillicuddy, 2002).
Finally, it is also worth noting that anticyclonic mesoscales eddies are capable of trapping near-inertial
energy waves in the ocean  (Kunze 1985, Danioux et al. 2008, Koszalka et al. 2010, Pallas-Sanz et al.,
2016) where they produce vertical recirculation patterns (Zhong and Bracco, 2013). Even if, some of
these dynamical aspects are partially resolved at 1/12° horizontal resolution, higher resolutions
simulations are necessary to correctly assess their specific impact.

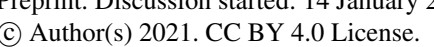
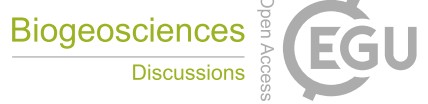

**Acknowledgments:** Research funded by the National Council of Science and Technology of Mexico –
Mexican Ministry of Energy – Hydrocarbon Trust, project 201441. This is a contribution of the Gulf of
Mexico Research Consortium (CIGoM). We acknowledge the provision of supercomputing facilities
by CICESE.



***APPENDIX A: CHL/C-biomass ratio and ecosystem structure***
[CHL] is widely used as a proxy for phytosynthetic biomass (Strickland, 1965; Cullen, 1982).
However, in addition to depend on phytoplankton concentration, it is also affected by several other
factors mainly produced by intracellular physiological mechanisms (Geider, 1987). In particular,
photoacclimation processes have been proved to be determinant to explain $[CHL]_{surf}$ variability in
oligotrophic areas (Mignot et al. 2014). In the GoM open-waters, this issue was specifically addressed
at a basin scale in Pasqueron de Fommervault et al. (2017) considering in-situ particulate
backscattering measurements and in Damien et al. (2018) from modeling tools. They both reach the
same conclusion: $[CHL]_{tot}$ variability provides a reasonably good estimate of the total C-biomass
variability ($[PHY]_{tot}$).
This is confirmed by the small amplitude of the seasonal cycle of the ratio $[CHL]_{tot}/[PHY]_{tot}$ in
the background GoM (0.256 +/- 0.004 g·mol$^{-1}$ averaged throughout the year, Fig A1). In the LCEs
core, this statement is still valid but must be qualified, since the ratio $[CHL]_{tot}/[PHY]_{tot}$ presents small
but significant changes through the year (Fig A1.a). It is around 0.24 g·mol$^{-1}$ from March to November
and increases sharply in December to reach about 0.32 g·mol$^{-1}$ in January and February. As a result, in
winter, the photoacclimation mechanism accounts for ~25% of the total $[CHL]_{tot}$ increase (the
remaining part being an effective phytoplankton biomass increase). In summer, the ratio
$[CHL]_{tot}/[PHY]_{tot}$ is slightly lower in the LCEs core compared to the background GoM. As a
consequence, the $[CHL]_{tot}$ negative anomaly associated with LCEs core does not necessarily translate
into a $[PHY]_{tot}$ negative anomaly.
Overall in the GoM open-waters, there is a dominance of the small-size phytoplankton over the
large-size class in proportion closed to 80%-20% (Linacre et al., 2015). Although the modeled





ecosystem structure is relatively simple, this typical community size structure is well reproduced by
GOLFO12-PISCES (Fig A1.c and A1.d), that also suggests a shift in the ecosystem structure in winter.
The different response among size classes results from the enhancement of nutrient vertical flux. The
role of "secondary" nutrient in this change in the community composition must not be overlooked also,
in particular for diatoms (accounted in the model's large-size group) since they also uptake on silicate
(Benitez-Nelson et al., 2007). Moreover, GOLFO12-PISCES exhibits a modulation of the ecosystem
structure by LCEs. The dominance of small-size phytoplankton is slightly more marked in summer and
the winter shift is stronger in the LCEs core.

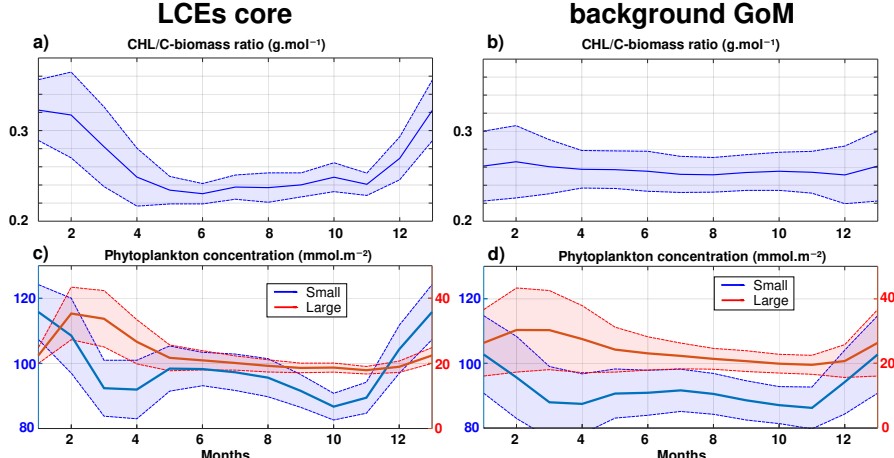

**Figure A1: Climatological seasonal cycles of (a and b) the CHL/C-biomass ratio and (c and d) the vertically integrated content of**
**phytoplankton concentration (small size in blue, large size in red). The left panels (a and c) refer to the time series in the LCEs**
**core (r < 50 km) whereas the right panels (b and d) refer to the time series in the background GoM (r > 200 km). For each**
**average cycle, the average value is shown (full line) along with its variability (+/- 1 standard deviation relative to the mean, dashed**
**lines).**





***APPENDIX B : Nitrate budget at a seasonal scale***

Nutrients availability in the euphotic layer is a key mechanism to trigger biomass increase in

LCEs. The processes driving the seasonality of nutrient concentrations are here investigated diagnosing
the different contributions to nitrate concentrations (hereafter [$NO_3$]) variability. The goal is to confirm
the vertical transport of nutrients and quantify the budget in order to determine the driving mechanisms.
The analysis is restricted to nitrate concentrations, considered as the main limiting factor for large size-
class phytoplankton growth in the GoM (Myers et al., 1981; Turner et al., 2006), although phosphates
and silicates are also modeled. We do not exclude that phosphates or silicates could also play a
significant role. In cylindrical coordinates, the [$NO_3$] equation reads:
$$\frac{\partial NO_3}{\partial t} = \underbrace{-V_r \frac{\partial NO_3}{\partial r}}_{radial\ advection} - \underbrace{\frac{V_\theta}{r}\frac{\partial NO_3}{\partial \theta}}_{azimuthal\ advection} - \underbrace{V_z \frac{\partial NO_3}{\partial z}}_{vertical\ advection} + \underbrace{\frac{D_l}{r}\frac{\partial}{\partial r}\left(r\frac{\partial NO_3}{\partial r}\right) + \frac{D_l}{r^2}\frac{\partial^2 NO_3}{\partial \theta^2}}_{lateral\ diffusion}$$
$$\underbrace{+\frac{\partial}{\partial z}\left(K_z \frac{\partial NO_3}{\partial z}\right)}_{vertical\ diffusion} + \underbrace{SMS}_{Source\ menus\ sink} + Asselin$$

Basically, this is a 3D advection-diffusion equation with added "sources and sinks" terms, namely
biogeochemical release and uptake rates. One must include also an "Asselin term", a modeling artifact
due to the Asselin time filtering. We focus on the seasonal cycle of three particular trend terms: the
vertical mixing (Fig B1.a and B1.b), the vertical advection (Fig B1.c and B1.d) and a "source menus
sink" term (Fig B1.e B1.f).

[$NO_3$] variations from vertical dynamics are mainly positive, especially in the first 100 m of the

water column. This traduces in year-round $NO_3$ source driven by physical processes. By contrast,
biogeochemical processes consume $NO_3$ in the upper layer to sustain the primary production (Fig B1.e
and B1.f). In the sub-surface layer (~ below the isoline on which nitrate concentration is equal to 2
mmol.m$^{-3}$), the process of nitrification constitutes a biological source of [$NO_3$]. To first order, this





represents the global functioning of the ecosystem, valid in both fields and throughout the year.
However, the seasonal cycle strongly influence the magnitude of these trend terms, in particular in the
LCE core.

In winter, from December to February, vertical advective and diffusive motions produce an

increase of [NO$_3$] within the mixed layer. This tendency consists in an advective entrainment resulting
from the deepening of the mixed layer which mainly acts to increase [NO$_3$] at the base of the mixed
layer (Fig B1.c and B1.d) and vertical mixing which redistributes vertically the nutrients and tends to
homogenize [NO$_3$] in the mixed layer (Fig B1.a and B1.b). The winter [NO$_3$] increase is most important
in the LCE core at the base of the mixed layer (~ + 6.5.10$^{-7}$ mmol·m$^{-3}$·d$^{-1}$, nearly 3 times larger than in
the background GoM), attesting here a preferential NO$_3$ uplift due to deeper convection. Integrated
over the mixed layer, the winter vertical fluxes produce [NO$_3$] enhancement of ~ 2.4.10$^{-5}$ mmol·m$^{-2}$·d$^{-1}$
in the eddy core whereas it is only of ~ 1.6.10$^{-5}$ mmol·m$^{-2}$·d$^{-1}$ in the background GoM. This also
explains why, on average, the density/nitrate relation differs in the LCEs core (Fig 5.e). In response, the
[NO$_3$] tendency due to biogeochemical processes indicates an increase of the [NO$_3$] uptake. This
increase is about 1.5 times larger in the core (~ - 1.3.10$^{-3}$ mmol·m$^{-2}$·d$^{-1}$ integrated over the mixed layer)
than in the background GoM (~ - 0.9.10$^{-3}$ mmol·m$^{-2}$·d$^{-1}$). Knowing that it feeds biomass production, this
[NO$_3$] loss is consistent with the primary production peak in winter (Fig 9.e and 9.f).

In summer, [NO$_3$] variations due to vertical processes are smaller than in winter. They are also

weaker in the LCEs core upper layer (almost nil in the 0-50m layer) compared to the background GoM,
consistent with a deeper NO$_3$ pool and a shallow mixer layer. In the eddy core, one can assume that the
NO$_3$ vertical supply is entirely consumed before reaching 50m. Below 50m, vertical [NO$_3$] diffusive
trends are consistently more important in the background GoM, in agreement with a steeper nitracline
(Fig 5.e). In contrast, vertical [NO$_3$] advective trends in the eddy core are similar to or can eventually



exceed the trends in the background GoM (as in September and October for example). This confirms a
pumping mechanism to sustain primary production in summer within the eddy core (section V.4) The
biogeochemical activity related to [NO₃] variations is also less intense in summer compared to winter.
The depth of maximum [NO₃] uptake is located just above the DCM and [NO₃] release below. The loss
of [NO₃] is about twice larger in the background GoM ($\sim -0.9.10^{-7}$ mmol·m⁻³·d⁻¹) than in the LCEs core
($\sim -0.5.10^{-7}$ mmol·m⁻³·d⁻¹). It is noteworthy that the biogeochemical [NO₃] source term, namely the
nitrification rate, is really low within the eddy core.

To close this analysis of the [NO₃] budget, it must be said that lateral diffusion and Asselin

tendencies are marginal terms compared to the others. Horizontal advection is of the same order of
magnitude as the vertical terms and mainly acts to redistribute horizontally the NO₃ vertically moved
(see supplementary material 1).

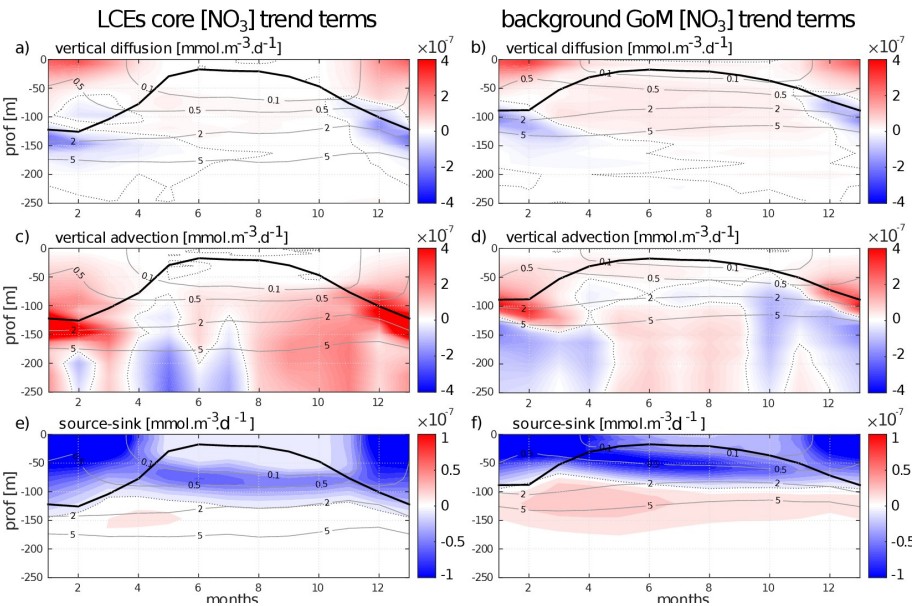

**Figure B1:** *Seasonal cycle of nitrate trend terms in the (left column) LCEs core and in the (right column) background GoM. The trend induced by (a and b) vertical mixing, the (c and d) vertical advection and the (e and f) biogeochemical source menus sink are represented. Isopycnals anomalies (gray contours) and the depth of the mixed layer (black line) are superimposed.*





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
