# Peer review of "Do Loop Current Eddies stimulate productivity in the Gulf of Mexico?"

_Biogeosciences, 2020_

## Referee Comment (RC2)

**General comments**

This study is to investigate the biological response within the Loop Current Eddies (LCEs), which are anti-cyclonic eddies in the Gulf of Mexico (GOM). Based on the satellite data in combined with a 3D coupled physical-biological model, the authors identify the positive anomalies of vertically integrated chlorophyll (chltot) within the core of LCEs in relative to the background GOM waters in the winter, which is, however, not reflected by the surface chlorophyll. In addition, the authors attribute this positive anomaly to the winter mixing within the core of LCEs.

In general, this study is very interesting, and its results can have important implications for the global carbon cycle. However, there remain some issues that by being addressed, the manuscript can be improved.

First, given that the winter increase of chlorophyll within the core of LCEs cannot be detected from the surface, I am concerned with the model performance of biological properties below the surface. The authors mention in the introduction that they will pay particular attention to the validation of eddy structures and surface chlorophyll. However, I would suggest more validation of subsurface biological properties inside and/or outside the LCEs. As I know, there are six autonomous floats which were deployed from 2010 to 2015 in the Gulf of Mexico. The authors can use these float measurements to do some model-data comparisons as they did in their previous paper (Damien et al., 2018) and to support their main conclusion, i.e. the positive anomaly of winter chltot within the core of LCEs. From my standpoint, this is necessary. For instance, model results show the nearly monotonically decreasing patterns of chlorophyll along the vertical direction in winter (Figure 6), which contrasts with the summer patterns with a distinct deep chlorophyll maximum (DCM). The winter chltot is higher because of the deeper inflection point and homogenized layer within the core of LCEs. However, based on the in-situ observations collected from autonomous floats in the Gulf of Mexico, the DCM is distinct throughout the whole year with the depth around 70-100m (Fommervault et al., 2017; Green et al., 2014). Model validation results (Figure 3 in Damien et al. (2018)) also show that this coupled model fails to reproduce the observed DCM in the winter. This could be a result of using suboptimal values of key biological parameters. Due to this model's weakness, the authors should be more careful about their results. Is it possible that the vertical profiles of chlorophyll respond to the LCEs in a similar way as they do in the summer, e.g., the deeper DCM and lower chltot? The authors should justify whether this model deficiency will change their main conclusions.

Second, some topics are not discussed comprehensively, making it look like a half-cooked product. For instance, the authors use salinity as a tracer to explain the eddy trapping mechanism. I really like it. However, there is no further discussion on its roles in the biological properties. Is the eddy trapping mechanism important for the positive anomalies in the core of winter LCEs? Is the positive anomaly produced locally within the LCEs or trapped from their original places during the eddies' formation? Based on the model results (Figure 8b, also 19 Line 323-328), the preferential increase of chltot within the winter LCEs is not observed before shedding and little differencse in chltot exist between the eddy center and background waters, which seems to support that the positive anomaly is produced locally. However, this behavior is largely determined by the poorly constrained open boundary conditions. Therefore, I would suggest the authors to complete this discussion based on their model results and float profiling observations.

Another example is in Section IV.4. The authors suggest that in the summer, the Ekman pumping within the LCEs can provide additional NO3 to sustain a comparable level of new primary productivity with the background waters. However, they don't explain the lower values of regenerated primary productivity, which determines the negative anomalies of chltot within the eddy. Which mesoscale mechanism is responsible? Why the new and regenerated primary productivity respond to the LCEs differently?

**Specific comments**

P7 Line 127-128: Could the authors explain more explicitly why a shallow detection depth can maximize the accuracy?

P9 Line 162: The authors seem to mix up the chlorophyll anomaly (in unit of mg m-3) and its normalized one (unitless). Based on their definition of normalized chlorophyll anomaly [CHL]', it should be unitless. However, they use chlorophyll anomaly almost throughout the whole manuscript without any definition (e.g. P11-12 Line195-200, Figure 7a, b). Based on the unit, I guess it might be calculated as $[CHL] - \overline{[CHL]}$. The authors should be clearer about it.

P12 Line 213-215: I can't see this paragraph because it is covered by the Figure 4

P23 Line 386-387: What's the definition of euphotic zone in this study. No figures show where the euphotic zone is.

P24 Figure 10. This figure is used to illustrate that in the winter of LCEs, the mixed layer is closer to the nitracline. However, it shows the results in summer (please see the figure caption).

Section IV.3: This subsection is not discussion. It should be in Results section.
Section IV.3: The grazing rate looks very important. What is the role of grazing rate in the positive anomalies of chltot within the core of winter LCEs? This top-down perspective will be interesting.

Section IV.4: It is unfair to compare the amplitude of annual averaged Ekman pumping with the deepening rate of mixed layer in the winter. What's the seasonal variability of the Ekman pumping?

P26 Line 449: Does this sentence mean that the vertical transport is a net effect of eddy pumping (downwelling in the LCEs) and eddy-wind interaction (upwelling in the LCEs)?

P28 Line 470: Does it means 0.06±0.13 m/day, or from +0.06 to -0.13 mg/day?

P28 Line 471-472: Can the authors refer to a figure which shows upwelling of isopycnals within the LCEs

P29 Line 97: As one of main conclusions, the authors never show anything about phytoplankton. As they mentioned before, the changes of chlorophyll can be a result of either the real change of phytoplankton or the photoacclimation. The authors should provide some results about the phytoplankton.

**Reference:**

Fommervault, O. P. De, Perez-brunius, P., Damien, P., Camacho-ibar, V. F. and Sheinbaum, J.: Temporal variability of chlorophyll distribution in the Gulf of Mexico: bio-optical data from profiling floats, Biogeosciences, 14, 5647–5662, doi:10.5194/bg-14-5647-2017, 2017.

Green, R. E., Bower, A. S. and Lugo-Fernandez, A.: First Autonomous Bio-Optical Profiling Float in the Gulf of Mexico Reveals Dynamic Biogeochemistry in Deep Waters, PLoS ONE, 9(7), 1–9, doi:10.1371/journal.pone.0101658, 2014.

---

## Author Comment (AC1)

Dear Dr. Zuo Xue,

Thank you for giving us the opportunity to submit a revised draft of our manuscript. We appreciate the effort that you have dedicated to providing this valuable feedback.

**Reviewer :** *Overall, I only have one suggestion-since the authors argue that the winter total PP is correlated not only eddy activity but also vertical mixing and the interaction between wind and the LCEs, why not carry out some sensitivity tests by manipulating the strength of wind-induced mixing in winter to support such hypothesis?*

**Authors :** We understand that your main concern regards the sensitivity of the primary production response to mixing within the LCE core. In a sense, this is what we did in the manuscript at the seasonal scale showing that the increase mixing in winter is associated with the increase in primary production. This is coherent with the seasonal and mesoscale focus of the study. For finer details, we could, following your suggestion, carry out a set of numerical runs with varying atmospheric forcing and/or mixing parametrization and look at the primary production response. However, this approach would be pretty heavy to implement since it would require several additional simulations over periods of time long enough to build robust LCE composites. It would also take out this study from its realistic context.

To derive this suggestion using the simulation described in the manuscript, we can argue that mixing presents significant variations on time scales ranging from one day to one week. Thus we can look at the [CHL] response to these variations within LCEs. This requires abandoning the methodology based on the seasonal composite analysis followed in the manuscript to adopt the Lagrangian point of view of each individual LCE. Figure R1 shows the time-series of surface density and chlorophyll concentration in the center of one LCE. A surface density increase can be interpreted as a proxy for mixing. It is clear that [CHL] shows variations at high frequency and there is indeed a correlation between mixing and a chlorophyll increase at high frequency in winter (yellow band). It is however not as robust as at seasonal scale since other drivers might become important at higher frequencies. Moreover, 5-day average outputs constitute probably an important limitation to study high-frequency processes.

Given these limitations, we agree with the reviewer that proper sensitivity test involving a set of new simulations would be helpful. However, we believe that expanding our dataset is hardly feasible, given the costs involved and would not significantly support our argument given the seasonal focus of the study.

Following is a point-by-point response to the reviewer' minor suggestions. The spelling and grammatical suggestions provided were incorporated. We have highlighted the changes within the manuscript.

**Reviewer :** *Line 105, details about the boundary and atmospheric forcing is needed, as well as rivers, although similar to Damien et al. 2018*
**Authors :** More details and references were added (lines 108-110).

**R:** *Line 205, what does "zonal" mean here?*
**A :** Zonal here means "along latitudes". We rephrased using "westward" to avoid confusion.

**R:** *208-212, here you mean the model results, so try not to use "images"*
**A :** We used "model outputs" instead (lines 215-222).

**R:** *Line 279-287 & Figure 6. Winter should be DJF and summer should be JJA, right?*
**A :** We choose to define the composite averaged over the months of January and February as "Winter" and the one averaged over "July and August" as "Summer" (c.f. lines 144-146).

**R:** *Line 323-328, very good figure 8. needs more words for the salinity as well*
**A :** Salinity is discussed in the discussion section IV.1 dealing with eddy-trapping mechanism

[Figure]

**Figure R1 : time-series of (upper panel) surface density and vertically integrated chlorophyll, and (lower panel) chlorophyll profile (in mgCHL.m⁻³) at the center of an individual LCE. Black contours refer to density anomaly and the x-axis is labeled in months.**

**R:** *Line 363, what is an eroded salinity maximum?*
**A:** "Eroded" was used here as a metaphoric synonym for "diffused". We rephrased it as "lower salinity maximum" (line 373)

      We would like to thank the referee again for taking the time to review our manuscript.

**References :**

Damien, P., Pasqueron de Fommervault, O., Sheinbaum, J., Jouanno, J., Camacho-Ibar, V. F., & Duteil, O. (2018). Partitioning of the open waters of the Gulf of Mexico based on the seasonal and interannual variability of chlorophyll concentration. Journal of Geophysical Research: Oceans, 123(4), 2592-2614.

---

## Author Comment (AC2)

Dear Referee,

Thank you for taking the time to assess our manuscript. We appreciate the effort that you have dedicated to provide your valuable feedback and insightful comments. We address the concerns that you raised in this response. We have been able to incorporate changes to reflect most of your suggestions highlighting them within the manuscript.

**Reviewer :** *First, given that the winter increase of chlorophyll within the core of LCEs cannot be detected from the surface, I am concerned with the model performance of biological properties below the surface. The authors mention in the introduction that they will pay particular attention to the validation of eddy structures and surface chlorophyll. However, I would suggest more validation of subsurface biological properties inside and/or outside the LCEs. As I know, there are six autonomous floats which were deployed from 2010 to 2015 in the Gulf of Mexico. The authors can use these float measurements to do some model-data comparisons as they did in their previous paper (Damien et al., 2018) and to support their main conclusion, i.e. the positive anomaly of winter chltot within the core of LCEs. From my standpoint, this is necessary. For instance, model results show the nearly monotonically decreasing patterns of chlorophyll along the vertical direction in winter (Figure 6), which contrasts with the summer patterns with a distinct deep chlorophyll maximum (DCM). The winter chltot is higher because of the deeper inflection pointand homogenized layer within the core of LCEs. However, based on the in-situ observations collected from autonomous floats in the Gulf of Mexico, the DCM is distinct throughout the whole year with the depth around 70-100m (Fommervault et al., 2017; Green et al., 2014). Model validation results (Figure 3 in Damien et al. (2018)) also show that this coupled model fails to reproduce the observed DCM in the winter. This could be a result of using suboptimal values of key biological parameters. Due to this model's weakness, the authors should be more careful about their results. Is it possible that the vertical profiles of chlorophyll respond to the LCEs in a similar way as they do in the summer, e.g., the deeper DCM and lower chltot? The authors should justify whether this model deficiency will change their main conclusions.*

**Authors:** This concern regarding the model performance below the surface is valid and important. Indeed, validation is a crucial step of modeling studies that pretend to simulate realistic conditions. We carried out an extensive validation of the modeled properties in a previous paper (Damien et al., 2018), focusing on properties that are known to influence primary production and chlorophyll concentration: mixed layer depth (appendix B of Damien et al., 2018) and the depth and slope of the nutricline (appendix D of Damien et al., 2018). A novel aspect was to use in-situ observations collected from autonomous floats and published in Green et al. (2014) and  Fommervault et al. (2017) to validate not only the modeled surface chlorophyll concentration but also the chlorophyll vertical profile in the Gulf of Mexico.  To be able to reproduce it correctly, the parameters of the biogeochemical model were largely tuned (Appendix C of Damien et al., 2018) compared to the ones suitable for global simulations (Aumont et al., 2015).  Based on Figure 3 in Damien et al. (2018), you raise doubts regarding the ability of the coupled model to reproduce the chlorophyll profile, particularly in winter. It is true that the mean winter chlorophyll profile presents a DCM, although it is much less remarkable that in the other months of the year. However, the variability associated with this average profile is very large. Looking at individual profiles during winter (Fig. 3 in Fommervault et al. (2017)), it results from well-mixed profiles in which no DCM can be observed (e.g in December 2013 in Argo float number 3 and 6) and profiles presenting a DCM (e.g December 2012 in float 4). These "stratified" profiles let a DCM footprint on the average profile observed in winter. It is likely that the relatively coarse model resolution fails in reproducing a variability as intense as observed in winter. It is also likely that it would underestimate restratifiying processes of the upper layer that would favor profiles presenting a DCM. Moreover, the northern, and more specifically the northeastern, region of the GoM is

undersampled by the Argo floats while it is where more chlorophyll monotonically decreasing profile are likely to be observed (Damien et al., 2018). Having said that, and beside the bias in the shape of the chlorophyll profile in winter, the vertically integrated chlorophyll content and its low seasonal variability compare nicely. Thus, in Damien et al. (2018), we demonstrate the ability of the coupled physical-biogeochemical model to reproduce the main observed features of the GoM, at least at a basin and seasonal scale which was the main scope of Damien et al., 2018.

Regarding the mesoscale scope of the manuscript you revised, an additional validation of biological properties at mesoscale were required and we propose comparison with surface chlorophyll inferred form satellite (Fig. 1 and 2). We point it more clearly in the revised version (lines 287-288). We also paid attention to the dynamical structure of the modeled LCEs (Fig. 2), which give us confidence regarding the modeling mixed layer and the nutricline in LCEs. In the manuscript, we show that the chlorophyll content increase in winter is associated only with the LCE core (radius<50km, Fig. 7). However, validating the chlorophyll vertical structure within the LCE core is difficult due to the lack of in-situ data. Meunier et al. (2018) show that the LCEs' core is associated with a strong negative potential density anomaly with the isopycnal 1026 kg.m$^{-3}$ reaching at least 300m depth within the core. In the BioArgo float data, this condition is reached only once, between September and November 2015 for float number 4 (Fig. 3 in Fommervault et al. (2017)). This event is not in winter and not associated with a remarkable signal in integrated chlorophyll content (Fig. 4 in Fommervault et al. (2017)).

We acknowledge that modeled vertical profiles of chlorophyll within LCEs can present some biases. However, due to the extensive model validation we carried out in a previous study and the validation of the subsurface biogeochemical properties presented in this manuscript, we are confident that the seasonal variability of the integrated content of chlorophyll is realistic. Moreover, this behavior associated with anticyclonic mesoscale eddies in oligotrophic environments is not observed for the first time and has been reported in the literature (Dufois et al., 2016 for example).

Having said that, we thank you for pointing out this limitation and, even if we are confident with the model's results, we agree that the manuscript needed to put more the results in perspective of the modeling framework and associated inherent biases. Therefore, we added the following in the conclusion of the new version: "Although GOLFO12-PISCES provides results which were confronted to observations, biases are inherent to model and these results would require confirmation by sub-surface in-situ measurements within the core of LCEs." (lines 526-528).

**Reviewer :** *Second, some topics are not discussed comprehensively, making it look like a half-cooked product. For instance, the authors use salinity as a tracer to explain the eddy trapping mechanism. I really like it. However, there is no further discussion on its roles in the biological properties. Is the eddy trapping mechanism important for the positive anomalies in the core of winter LCEs? Is the positive anomaly produced locally within the LCEs or trapped from their original places during the eddies' formation? Based on the model results (Figure 8b, also 19 Line 323-328), the preferential increase of chltot within the winter LCEs is not observed before shedding and little differences in chltot exist between the eddy center and background waters, which seems to support that the positive anomaly is produced locally. However, this behavior is largely determined by the poorly constrained open boundary conditions. Therefore, I would suggest the authors to complete this discussion based on their model results and float profiling observations.*

**Authors :** The second main point you address relates with unclear or limited elements in the discussion. Starting with the eddy trapping mechanism, we indeed use salinity to evidence the trapping mechanism. We agree that its role in the biological properties is not expressed clearly. In fact, the eddy

trapping mechanism implies that the winter $[CHL]_{tot}$ increase has to be driven by local processes. We thank the reviewer for pointing that this important statement, wich is a milestone in the discussion, was missing. We added to the revised manuscript: "Since no significant $[CHL]_{tot}$ seasonal variability is reported in the Western Caribbean Sea (Fig. 8), the biogeochemical behavior in the LCEs core has then to be driven by local processes with low influence of horizontal advective process from the ring or of the Caribbean waters trapped during the LCEs formation." (lines 382-385).

The problem of open boundary conditions and how they may drive the model results is unfortunately an inescapable issue of regional modeling studies. However, their influence can be limited considering several precaution followed in this study. First, the study focus on a region relatively distant from the boundaries, especially the ones located upstream in the Caribbean Sea, so that the model can developed its own dynamic and biogeochemical cycles. Then, a particular attention was paid to the condition applied at these boundaries. In our case, we fitted the vertical distribution from the World Ocean Atlas observation database or the global standard configuration ORCA2 at the boundary location to the density profile applied (Damien et al., 2018). This method was proved to produce nutrient concentrations inside the Gulf of Mexico in good agreement with observations (see Annexe D of Damien et al., 2018).

**Reviewer :** *Another example is in Section IV.4. The authors suggest that in the summer, the Ekman pumping within the LCEs can provide additional NO3 to sustain a comparable level of new primary productivity with the background waters. However, they don't explain the lower values of regenerated primary productivity, which determines the negative anomalies of chltot within the eddy. Which mesoscale mechanism is responsible? Why the new and regenerated primary productivity respond to the LCEs differently?*

**Authors :** With respect to the different components of primary production in summer, we indeed focus on the new primary production in section IV.4 since we found very interesting that it has similar rates in the LCEs core andin the GoM background while nitrate are found much deeper. Thank you for pointing that we did not provide any explanation for the lower value of regenerated primary production. We observe that the grazing rate is lower inside the LCE compared to the GoM background during summer (Fig. 9. c.d.). Since grazing is known to be a major source of recycled nutrients in the euphotic zone (Sherr and Sherr, 2002), it explains the lower regenerated primary production. We can also add that production of organic matter occurs in a deeper layer within the LCEs core compared to the background GoM (Fig. B1,e,f). It is then more likely exported out of the euphotic layer in the form of settling particle, leading to lower remineralization rates in the upper layers and less available $NH_4$ to feed regenerated production. We have accordingly updated the first paragraph of section IV.4 to reflect this point (lines 461-463) : "Since grazing is known to be a major source of recycled nutrients in the euphotic zone (Sherr and Sherr, 2002), the lower grazing rate inside the LCE during summer (Fig. 9. c. d.) likely explains this lower regenerated production."

Here follows a point-by-point response to the specific comments :

**Reviewer :** *P7 Line 127-128: Could the authors explain more explicitly why a shallow detection depth can maximize the accuracy?*

**Authors :** Applying the algorithm where velocities have larger magnitude usually facilitate the detection and tracking since vorticity tends to be more intense too. Having said that, this detail is probably not important and we have removed it from the revised manuscript.

**R :** *P9 Line 162: The authors seem to mix up the chlorophyll anomaly (in unit of mg m-3) and its normalized one (unitless). Based on their definition of normalized chlorophyll anomaly [CHL]', it*

*should be unitless. However, they use chlorophyll anomaly almost throughout the whole manuscript without any definition (e.g. P11-12 Line195-200, Figure 7a, b).*

**A :** This observation is correct and we thank you for pointing this out. We use the chlorophyll anomaly almost through the whole manuscript and we hence provide its definition (lines 169-170). We used the normalized anomaly to perform the EOF analysis. We clarify this in the revised manuscript to avoid any confusion (line 320 and caption of figure 7).

**R :** *P12 Line 213-215: I can't see this paragraph because it is covered by the Figure 4*
**A :** We apologize for this error. We fixed it in the revised manuscript.

**R :** *P23 Line 386-387: What's the definition of euphotic zone in this study. No figures show where the euphotic zone is.*
**A :** Thank you for pointing this out. We have added the depth of the euphotic layer in the GoM according to Jolliff et al. (2008) and confirmed by Linacre et al. (2019) from in-situ measurements (line 166). These references show that the depth of the euphotic layer reaches between 120 and 150 m and that the mixed layer in the Gulf of Mexico does not exceed the euphotic layer, even in winter, implying that new primary production responds directly to an increased upward nutrient flux triggered by winter mixing.

**R :** *P24 Figure 10. This figure is used to illustrate that in the winter of LCEs, the mixed layer is closer to the nitracline. However, it shows the results in summer (please see the figure caption).*
**A :** An important point is raised here. We argue in the manuscript that an important driver of the depth reached by the mixed layer is the stratification of the water column before the winter mixing (line 408). A lower stratification in the pre-winter season, of which the summer columnar buoyancy as we define it is a good metric, would imply a deeper mixing. As a consequence, it is appropriate to show the columnar buoyancy in summer to argue that the LCEs core is conditioned for deeper mixing in winter. We added in the caption of figure 10 the clarification that "summer" corresponds to the pre-winter mixing season.

**R :** *Section IV.3: This subsection is not discussion. It should be in Results section.*
**A :** It is indeed true that this section provide substantial new results. However, since it is used to develop and discuss the biogeochemical driver of the chlorophyll variability (primary production and grazing), we find more appropriate to keep this organization.

**R :** *Section IV.3: The grazing rate looks very important. What is the role of grazing rate in the positive anomalies of chltot within the core of winter LCEs? This top-down perspective will be interesting.*
**A :** We agree that this aspect is interesting to explore.
The literature reports that the percentage of primary production grazed by microzooplankton varies between 50 and 77% (Calbet and Landry, 2004; Schmoker et al., 2013). An averaged 90% of the total growth consumed by grazers is indeed more important. However, PISCES model includes grazing by mesozooplankton (Aumont et al., 2015) which is still not well quantified.
Since the grazing rate shows a similar seasonal cycle and similar magnitudes relative to the primary production within the LCE core and in the GoM background, its role in the positive anomaly of chlorophyll is likely secondary compared to the primary production increase. However, the zooplankton increase (and the associated grazing) is not responding in a linear way with primary production. In February, the difference between primary production and grazing rate is larger in the core than in the GoM background (Fig. 9.c). It participates then in the larger net primary production and enhancing the phytoplankton concentration in the LCE core compared to the background. This top-down perspective is actually interesting and discussed in a new paragraph at the end of section IV.3:

"The pressure exerted by zooplankton grazers varies seasonally  (Fig 9.c .d). It shows a similar seasonal cycle in the LCEs core and in the background GoM. On average, ~ 90% of the total growth is consumed by grazers, reaching the highest impact in March, just one month after the peak season of the $PP_{tot}$ in both areas. In February the difference between the primary production and the grazing rate is larger in the LCEs core than in the GoM background (Fig. 9.c), leading to an enhanced net primary production. Considering the ecosystem from a "top-down" perspective, the grazing rate also participates then in enhancing $[CHL]_{tot}$ within the LCEs core compared to the background."

**R :** *Section IV.4: It is unfair to compare the amplitude of annual averaged Ekman pumping with the deepening rate of mixed layer in the winter. What's the seasonal variability of the Ekman pumping?*
**A :** We computed the time-series of the Ekman pumping estimated with the wind magnitude over the LCEs. Even if the wind shows larger magnitudes in winter, it is also associated with a large variability (de Velasco and Winant, 1996). As a consequence, the variability of Ekman pumping was also found large. We cannot identify a robust seasonal seasonal cycle which would allow to define a summer and a winter Ekman pumping. However, we think that the scaling we propose remains pretty robust since there is a difference of one order magnitude between winter mixing and Ekman pumping. This is true even when we consider the pumping associated with a wind among the strongest observed in the GoM and occurring  in winter (about 7.5 m·s$^{-1}$ usually due cold fronts, Passalacqua et al., 2016). We rephrased lines 504-505 in order to be more specific.

**R :** *P26 Line 449: Does this sentence mean that the vertical transport is a net effect of eddy pumping (downwelling in the LCEs) and eddy-wind interaction (upwelling in the LCEs)?*
**A :** We mean that these two mechanisms contribute to upwelling in the LCEs. During its formation and, as the rotational velocities increase, the eddy pumping in anticyclone is directed downward. However, as an LCE detaches form the Loop Current, its rotational velocities tend to decay and eddy pumping is then directed upward (Flierl, G., & McGillicuddy, D. J., 2002). We mention this at lines (473-474).

**R :** *P28 Line 470: Does it means 0.06±0.13 m/day, or from +0.06 to -0.13 mg/day?*
**A :** The formulation we used is indeed very misleading and we apologize for this. We mean "range from  0.06 to 0.13". We rephrased accordingly in the revised manuscript (lines 491).

**R :** *P28 Line 471-472: Can the authors refer to a figure which shows upwelling of isopycnals within the LCEs*
**A :** We remove the mention of isopycnals form this sentence to avoid misunderstandings and refer to Fig. 11 (line 493).

**R :** *P29 Line 97: As one of main conclusions, the authors never show anything about phytoplankton. As they mentioned before, the changes of chlorophyll can be a result of either the real change of phytoplankton or the photoacclimation. The authors should provide some results about the phytoplankton.*
**A :** We acknowledge the limitation of limitation of chlorophyll as a proxy for phytoplankton in the manuscript and provide in appendix A the chlorophyll over carbon ratio.

We would like to thank the referee again for taking the time to review our manuscript.

**References :**

Aumont, O., Ethé, C., Tagliabue, A., Bopp, L., & Gehlen, M. (2015). PISCES-v2: An ocean biogeochemical model for carbon and ecosystem studies. Geoscientific Model Development, 8(8), 2465–2513.

Calbet, A., & Landry, M. R. (2004). Phytoplankton growth, microzooplankton grazing, and carbon cycling in marine systems. *Limnology and Oceanography*, *49*(1), 51-57.

Damien, P., Pasqueron de Fommervault, O., Sheinbaum, J., Jouanno, J., Camacho-Ibar, V. F., & Duteil, O. (2018). Partitioning of the open waters of the Gulf of Mexico based on the seasonal and interannual variability of chlorophyll concentration. *Journal of Geophysical Research: Oceans*, *123*(4), 2592-2614.

Dufois, F., Hardman-Mountford, N. J., Greenwood, J., Richardson, A. J., Feng, M., & Matear, R. J. (2016). Anticyclonic eddies are more productive than cyclonic eddies in subtropical gyres because of winter mixing. Science advances, 2(5), e1600282.

Flierl, G., & McGillicuddy, D. J. (2002). Mesoscale and submesoscale physical-biological interactions. *The sea*, *12*, 113-185.

Green, R. E., Bower, A. S., & Lugo-Fernández, A. (2014). First autonomous bio-optical profiling float in the Gulf of Mexico reveals dynamic biogeochemistry in deep waters. *PloS one*, *9*(7), e101658.

Jolliff, J. K., Kindle, J. C., Penta, B., Helber, R., Lee, Z., Shulman, I., Arnone, R., and Rowley, C. D., (2008). On the relationship between satellite-estimated bio-optical and thermal properties in the Gulf of Mexico, J. Geophys. Res., 113, G1, https://doi.org/10.1029/2006JG000373

Linacre, L., Durazo, R., Camacho-Ibar, V. F., Selph, K. E., Lara-Lara, J. R., Mirabal-Gómez, U., ... & Sidón-Ceseña, K. (2019). Picoplankton Carbon Biomass Assessments and Distribution of Prochlorococcus Ecotypes Linked to Loop Current Eddies During Summer in the Southern Gulf of Mexico. *Journal of Geophysical Research: Oceans*, *124*(11), 8342-8359.

Meunier, T., Pallás-Sanz, E., Tenreiro, M., Portela, E., Ochoa, J., Ruiz-Angulo, A., & Cusí, S. (2018). The vertical structure of a Loop Current Eddy. *Journal of Geophysical Research: Oceans*, *123*(9), 6070-6090.

Pasqueron de Fommervault, O., Perez-Brunius, P., Damien, P., Camacho-Ibar, V. F., & Sheinbaum, J. (2017). Temporal variability of chlorophyll distribution in the Gulf of Mexico: bio-optical data from profiling floats. *Biogeosciences*, *14*(24), 5647-5662.

Passalacqua, G. A., Sheinbaum, J., & Martinez, J. A. (2016). Sea surface temperature influence on a winter cold front position and propagation: Air-sea interactions of the 'Nortes' winds in the Gulf of Mexico. Atmospheric Science Letters, 17(5), 302–307.

Schmoker, Claire, Santiago Hernández-León, and Albert Calbet. "Microzooplankton grazing in the oceans: impacts, data variability, knowledge gaps and future directions." *Journal of Plankton Research* 35.4 (2013): 691-706.

Sherr, E. B., & Sherr, B. F. (2002). Significance of predation by protists in aquatic microbial food webs. *Antonie van Leeuwenhoek*, *81*(1), 293-308.

de Velasco, G. G., & Winant, C. D. (1996). Seasonal patterns of wind stress and wind stress curl over the Gulf of Mexico. *Journal of Geophysical Research: Oceans, 101*(C8), 18127-18140.

---

## Referee Report (RR1)

**General comments:**

The authors made significant improvements in this revised manuscript and responded to most comments. However, they failed to address some of my comments. In addition, there are still some flaws in the revised manuscript.

**Specific comments:**

I would suggest the authors to also include the open boundary conditions of biological component.

P9 Line158 of the revision-tracked manuscript: The stratification would result in the shallow mixed layer.

P9 Line172 of the revision-tracked manuscript: Please give the definition of the euphotic layer: the depth where the light intensity is 1% of the surface?

P16 Line276-277 of the revision-tracked manuscript: This sentence is confusing. I would suggest to change the 'associated with' into 'accompanied by'.

P17 Line293-294 of the revision-tracked manuscript: The contrasted seasonal response to the LCEs cannot be recognized by the surface chlorophyll. This is one of the main conclusions of this manuscript.

P17 Line300-304 of the revision-tracked manuscript: As in my first general comment, I am concerned about the biological model performance in the subsurface and the potential influence of this model weakness on the main conclusion: the increased winter chltot within the core of the LCEs. As the authors mentioned in their response, the vertical profiles of chlorophyll vary a lot in the winter: some individual profiles are well mixed without the DCM while others are 'stratified' with the distinct DCM. Based on Damien et al 2018 to which the authors referred their model validation, the averaged winter profile shows a distinct DCM at about 60m with the $[Chl]_{DCM}$ ~50% higher than the $[Chl]_{surf}$. All of these clues suggests the significance of 'stratified' chlorophyll profiles in the winter of Gulf of Mexico. The failure of the biological model to reproduce these important 'stratified' profiles may have large influence on the results. At least, the results of this paper do not apply to these 'stratified' winter profiles. I am not asking the authors to re-run the model or to accept my opinion, but I hope that the authors can fully discuss it in their manuscript and be cautious about their conclusions.

P24 Line 405-407 of the revision-tracked manuscript: Where is the euphotic layer? Could the authors plot it along with the nitracline? The authors mentioned it earlier in their revised manuscript that the euphotic layer can reach between 120 and 150 meters in the Gulf of Mexico. If it applies here as well, the nitracline is still within the euphotic layer.

P26 Line 448-452 of the revision-tracked manuscript: Please refer to Figure 9

P27 Line 466-467 of the revision-tracked manuscript: It is hard to see the differences in the decoupling of production and grazing between the eddy core and background GOM (e.g.

GRZtot:PPtot both about 0.95 on February) from the figure. I would suggest the authors to provide the mean values and the standard deviation, and to reduce the y-axis scales of the figure.

P27 Line 472 of the revision-tracked manuscript: The major source of recycled nutrients should be the remineralization.

P28 Line 474-477 of the revision-tracked manuscript: How did the authors draw this conclusion from Figure b1 which show the biological source and sinks of NO3? Could the authors explain it clearer.

P30 Line 511-524 of the revision-tracked manuscript: This paragraph discussed the relative importance of two mechanisms (eddy pumping and eddy-wind interaction) in the winter and should not be in this subsection (Section IV. 4 How to explain summer productivity). I can understand that it is an extension of the discussion on two mechanisms, but I hope that the authors can re-organize it better.

Maybe because I am not a physical oceanographer, what is the definition of eddy-Ekman pumping? Is it Eddy-wind interactions? Please make it clearer.

P30 Line 519 of the revision-tracked manuscript: the word 'seasonal' is duplicated. Please remove one.

P34 Line 610 of the revision-tracked manuscript: I guess this should be 80%:20%, right?

Caption of Figure B1: 'minus', not 'menus'

---

## Author Response (AR2)

Dear Referee,

Thank you for taking the time to assess our revised manuscript. You raised several specific comments that we are addressing here after. We incorporated some changes to the manuscript to reflect your suggestions.

*Reviewer : I would suggest the authors to also include the open boundary conditions of biological component.*
**Authors:** We included the open boundary conditions used for the biogeochemical tracers in the section II.1: "The open-boundary conditions of biogeochemical tracers are prescribed from the World Ocean Atlas observation database (Garcia et al., 2010) for $NO_3$, $O_2$, Si, and $PO_4$, and from the global configuration ORCA2 (Aumont & Bopp, 2006) for DIC, DOC, Alkalinity, and Fe. The other state variables are forced arbitrary very small constant values." (P6 Line 107-110 of the revision-tracked manuscript).

*Reviewer : P9 Line158 of the revision-tracked manuscript: The stratification would result in the shallow mixed layer.*
**Authors:** We preferred avoiding any preliminary interpretation at this stage of the manuscript. We simply rephrased as follow : "The stratification of the water column is evaluated by the square of the buoyancy frequency" (P9 Line 157-158 of the revision-tracked manuscript).

*Reviewer : P9 Line172 of the revision-tracked manuscript: Please give the definition of the euphotic layer: the depth where the light intensity is 1% of the surface?*
**Authors:** We gave the metric used to define the euphotic layer : "The euphotic depth corresponds to 1% of the incoming photosynthetic active radiation at surface" (P9 Line 172-173 of the revision-tracked manuscript).

*Reviewer : P16 Line276-277 of the revision-tracked manuscript: This sentence is confusing. I would suggest to change the 'associated with' into 'accompanied by'.*
**Authors:** We thank the reviewer to point that the sentence was confusing and followed its suggestion (P16 Line 275 of the revision-tracked manuscript)

*Reviewer : P17 Line293-294 of the revision-tracked manuscript: The contrasted seasonal response to the LCEs cannot be recognized by the surface chlorophyll. This is one of the main conclusions of this manuscript.*
**Authors:** We agree that this can be interpreted as a contradiction with the main conclusion of the manuscript. To avoid any misunderstanding, this was deleted of the revised version (P17 Line 292-293 of the revision-tracked manuscript).

*Reviewer : P17 Line300-304 of the revision-tracked manuscript: As in my first general comment, I am concerned about the biological model performance in the subsurface and the potential influence of this model weakness on the main conclusion: the increased winter chltot within the core of the LCEs. As the authors mentioned in their response, the vertical profiles of chlorophyll vary a lot in the winter: some individual profiles are well mixed without the DCM while others are 'stratified' with the distinct DCM. Based on Damien et al 2018 to which the authors referred their model validation, the averaged winter profile shows a distinct DCM at about 60m with the [Chl] DCM ~50% higher than the [Chl] surf . All of these clues suggests the significance of 'stratified' chlorophyll profiles in the winter of Gulf of Mexico. The failure of the biological model to reproduce these important 'stratified' profiles may have large influence on the results. At least, the results of this paper do not apply to these 'stratified' winter*

*profiles. I am not asking the authors to re-run the model or to accept my opinion, but I hope that the authors can fully discuss it in their manuscript and be cautious about their conclusions.*
**Authors:** We understand the concern of the reviewer and we tried in our first response to demonstrate the robustness of our modeling results at a basin and mesoscale scales although acknowledging that modeled vertical profiles of chlorophyll can individually present some biases. In a revised version, we discuss more into details these biases and how they can affect the main conclusions of the study (P32 lines 546-553 of the revision-tracked manuscript).

*Reviewer : P24 Line 405-407 of the revision-tracked manuscript: Where is the euphotic layer? Could the authors plot it along with the nitracline? The authors mentioned it earlier in their revised manuscript that the euphotic layer can reach between 120 and 150 meters in the Gulf of Mexico. If it applies here as well, the nitracline is still within the euphotic layer.*
**Authors:** The base of the euphotic layer and the nitracline were added to the figure 9. The nitracline, defined as a nitrate concentration threshold in this study, is used as a practical metric of which relative position with the mixed layer depth controls the amount of nutrient injected into the surface layer in winter. The surface nitrate concentrations confirm that the relative position of the MLD and the nitracline controls the amount of nutrient supply to the surface in iwnter and consequently the $[CHL]_{tot}$ increase. As defined here, the nitracline is indeed still within the euphotic layer. In case we would want to investigate the nutrient flux through the base of the euphotic layer, a more restrictive threshold would be preferable to define the nitracline.

*Reviewer : P26 Line 448-452 of the revision-tracked manuscript: Please refer to Figure 9*
**Authors:** Done (*P26 Line 448* of the revision-tracked manuscript)

*Reviewer : P27 Line 466-467 of the revision-tracked manuscript: It is hard to see the differences in the decoupling of production and grazing between the eddy core and background GOM (e.g.GRZtot:PPtot both about 0.95 on February) from the figure. I would suggest the authors to provide the mean values and the standard deviation, and to reduce the y-axis scales of the figure.*
**Authors:** The mean values and standard deviation have been added to the text (P27 Line 363-364 of the revision-tracked manuscript). The y-axis of the figure can hardly be reduced if we do not want to cut off the variability envelope.

*Reviewer : P27 Line 472 of the revision-tracked manuscript: The major source of recycled nutrients should be the remineralization.*
**Authors:** Right. Remineralization is the major source of recycled nutrient and grazing feed remineralization by the production of dissolved and particulate organic matter. To avoid this ambiguity without going too much into details, we rephrased as followed : "Since grazing is known to be a major contributor of the recycling loop in the euphotic zone, …" (P27 Line 469-470 of the revision-tracked manuscript).

*Reviewer : P28 Line 474-477 of the revision-tracked manuscript: How did the authors draw this conclusion from Figure b1 which show the biological source and sinks of NO3? Could the authors explain it clearer.*
**Authors:** Regarding inorganic nutrient such as nitrate ($NO_3$), the production of organic matter (or primary production) expresses as a biogeochemical sink. In figure B1, we can see that the intense primary production rates in winter is associated with a strong consumption of nitrate. This $NO_3$ biogeochemical sink is larger and occurs on a thicker surface layer within the LCE core. In the revised manuscript, we explained it clearer: "In addition, the biogeochemical consumption of nitrate that foster

the production of organic matter occurs in a deeper layer within the LCEs core compared to the background GoM (Fig. B1. e. f.).” (P28 Line 472-473 of the revision-tracked manuscript)

*Reviewer : P30 Line 511-524 of the revision-tracked manuscript: This paragraph discussed the relative importance of two mechanisms (eddy pumping and eddy-wind interaction) in the winter and should not be in this subsection (Section IV. 4 How to explain summer productivity). I can understand that it is an extension of the discussion on two mechanisms, but I hope that the authors can re-organize it better. Maybe because I am not a physical oceanographer, what is the definition of eddy-Ekman pumping? Is it Eddy-wind interactions? Please make it clearer.*

**Authors:** We understand that the title of the section is the main reason of the reviewer confusion while reading this last paragraph.  He is right since it explicitly mentions "summer" while this last paragraph focus on the relative importance of two mechanisms in winter. In the revised version, we changed this section name to "Eddy-wind interactions" to keep it coherent with the content of the section. Eddy-Ekman pumping is eddy-wind interactions. We make it clearer in the text (P29 Line 491-492 of the revision-tracked manuscript)

*Reviewer : P30 Line 519 of the revision-tracked manuscript: the word 'seasonal' is duplicated. Please remove one.*

**Authors:** Done. We thanks the reviewer for reporting this typo.

*Reviewer : P34 Line 610 of the revision-tracked manuscript: I guess this should be 80%:20%, right? Caption of Figure B1: 'minus', not 'menus'*

**Authors:** It is 80%:20%. The text has been updated according to this notation. The error in the caption was corrected.